# Cloud heterogeneity on cloud and aerosol above cloud properties retrieved from simulated total and polarized reflectances

Céline Cornet[1], Laurent C-Labonnote[1], Fabien Waquet[1], Frédéric Szczap[2], Lucia Deaconu[1], Frédéric Parol[1], Claudine Vanbauce[1], François Thieuleux[1], Jérôme Riédi[1]

[1]Université Lille, CNRS, UMR 8518 - LOA - Laboratoire d'Optique Atmosphérique, F-59000 Lille, France
[2]Université Clermont Auvergne, CNRS, UMR 6016, Laboratoire de Météorologie Physique, F-63000 Clermont-Ferrand, France

*Correspondence to*: Céline Cornet (celine.cornet@univ-lille1.fr)

**Abstract.** Simulations of total and polarized cloud reflectance angular signatures such as the ones measured by the multi-angular and polarized radiometer POLDER3/PARASOL are used to evaluate cloud heterogeneity effects on cloud parameter retrievals. Effects on optical thickness, albedo, effective radius and variance of the cloud droplet size distribution and aerosol above cloud optical thickness are analyzed. Three different clouds having the same mean optical thicknesses were generated: the first one with a flat top, the second one with a bumpy top and the last one with a fractional cloud cover. At small scale (50 m), for oblique solar incidence, the illumination effects lead to higher total but also polarized reflectances. The polarized reflectances even reach values that cannot be predicted by the 1D homogeneous cloud assumption. At the POLDER scale (7 km x 7 km), the angular signature is modified by a combination of the plane-parallel bias and the shadowing and illumination effects. In order to quantify effects of cloud heterogeneity on operational products, we ran the POLDER operational algorithms on the simulated reflectances to retrieve the cloud optical thickness and albedo. Results show that the cloud optical thickness is greatly affected: biases can reach up to -70%, -50% or +40% for backward, nadir and forward viewing directions respectively. Concerning the albedo of the cloudy scenes, the errors are smaller, between -4.7% for solar incidence angle of 20° and up to about +8% for solar incidence angle of 60°. We also tested the heterogeneity effects on new algorithms that allow retrieving cloud droplet size distribution and cloud top pressures and also aerosol above clouds. Contrarily to the bi-spectral method, the retrieved cloud droplet size parameters are not significantly affected by the cloud heterogeneity, which proves to be a great advantage of using polarized measurements. However the cloud top pressure obtained from molecular scattering in the forward direction can be biased up to 120 hPa (around 1 km). Concerning the aerosol optical thickness (AOT) above cloud, the results are different depending on the available angular information. Above the fractional cloud, when only side scattering angles between 100 and 130° are available, the AOT is underestimated because of the plane-parallel bias. For solar zenith angle of 60°, on contrary, it is overestimated because the polarized reflectances are increased in forward directions.

# 1 Introduction

Cloud properties such as effective radius, optical thickness and albedo are key parameters for studies concerning cloud radiative effects and hydrological cycle of Earth climatic system. In the context of climate change, these properties may be modified and result in a feedback, the sign of which remains largely uncertain. In parallel, anthropogenic activities modify the aerosol loading in the atmosphere and consequently play an important role on cloud through the indirect radiative effects of aerosols (Twomey et al. 1977). In addition, absorbing aerosol above clouds can generate a positive direct radiative forcing (i.e. warming), that is currently not well quantified, and modify the properties of the below cloud layer (Chand et al., 2008; Costantino and Bréon, 2013; Wilcox, 2010).

Currently, several satellite radiometers use solar and infrared reflectances to infer cloud and aerosols above cloud parameters. Generally, cloud optical thickness (COT) and albedo are obtained from visible channels. Depending on instrument capabilities, the effective radius can be retrieved jointly with the optical thickness from a combination of visible and near-infrared measurements (Nakajima and King, 1990) as it is done in the operational algorithm of the Moderate Resolution Imaging Radiometer (MODIS Platnick et al., 2003). These parameters can also be retrieved separately from multi-viewing total and polarized measurements (Buriez et al., 1997a; Bréon and Goloub, 1998) as implemented for the optical thickness or under implementation for the effective radius with the Polarization and Directionality of the Earth's Reflectances Radiometer, (POLDER, Deschamps et al., 1994).

Concerning aerosols, spaceborne active instruments, such as the lidar CALIOP are dedicated tools to detect multi-layer situations and to retrieve Aerosol Above Cloud (AAC) properties (Hu et al., 2007; Chand et al., 2008; Young and Vaughan, 2009) and were used for climate studies (Zhang et al., 2016a). Passive measurements, that allows a larger global coverage, can also be used. An operational algorithm was developed to retrieve AAC scenes from the polarization measurements provided by the POLDER instrument onboard PARASOL (Waquet et al., 2009, 2013a) and was used to provide global analysis of the aerosol above clouds properties (Waquet et al., 2013b). Further, (Peers et al., 2015) combined total and polarized radiance measurements to retrieve the aerosol absorption above clouds. A color ratio technic was also developed to retrieve the AAC optical thickness and the corrected cloud optical thickness from total radiance measurements. This method was adapted to OMI UV measurements and MODIS multi-spectral measurements (Torres et al., 2011; Meyer et al., 2015)

For computation time and simplicity reasons, all these operational algorithms assume that clouds are flat, homogeneous and horizontally infinite, which is quite far from the reality. Numerous studies presented for example in Davis and Marshak (2010) or in Marshak and Davis (2005) showed that this assumption can lead to large errors on the retrieved cloud parameters. For example, the cloud optical thickness can be affected by the so-called plane-parallel bias induced by the sub-pixel heterogeneity and the non-linear relationship between reflectances and optical thickness. This bias usually leads to an effective optical thickness lower than the mean optical thickness (Cahalan, 1994; Szczap et al., 2000a). The sub-pixel optical thickness heterogeneity can also cause a positive bias on the mean effective radius retrieved following the bi-spectral technique (Szczap

et al., 2000b; Zhang et al., 2012), whereas the sub-pixel microphysical heterogeneity, not studied in this paper, leads on the contrary to an underestimation of the effective radius (Marshak et al., 2006). The bias on effective radius can thus be positive or negative depending on sub-pixel heterogeneity of the cloud optical thickness and effective radius (Zhang et al., 2016b).

In addition to the sub-pixel heterogeneity, Loeb and Davies (1996) detected an increase of the retrieved optical thickness from AVHRR (Advanced Very High Resolution) correlated with the solar zenith angle. Indeed, for oblique solar illumination, more energy is transmitting through the clouds along the cloud side (or bump). It leads to increase the upward reflectances. Consequently the cloud optical thickness retrieved under the homogeneous cloud assumption appears higher for tilted sun than for overhead sun. This effect is combined with angular effects, known as 3D effects which depend on the sensor viewing direction. Again, in the backward scattering direction, parts of the cloud sides are illuminated by the Sun and lead to a larger retrieved optical thickness value. Inversely, in viewing directions close to the forward scattering directions, some parts of the cloud are in the shadow resulting in smaller optical thickness or larger effective radius. This angular signature was observed on the retrieved cloud optical thickness by several radiometers such as AVHRR (Loeb and Coakley, 1998), MODIS (Varnai and Marshak, 2002) and POLDER (Buriez et al., 2001; Zeng et al., 2012).

Concerning Aerosol Above Cloud (AAC), intercomparisons of passive and active retrievals were performed for case studies (Jethva et al., 2013) and for global and multi-year data (Deaconu et al., 2017). The methods developed for passive instruments are all based on 1D calculations and, so, generally restricted to homogeneous cloudy pixels, for which the 3D effects are minimized. In case of aerosol retrieval in partial cloudy scenes, shadow, cloud enhancement of the clear areas by neighboring clouds can also modify the retrieved aerosol properties. Errors on the retrieved aerosol properties are in general dependent of the cloud distribution, optical thickness and spatial resolution (Stap et al., 2016a; Stap et al., 2016b).

Therefore, depending on the cloud heterogeneity, solar zenith angle and viewing geometry, cloud parameters (i.e. optical thickness and effective radius) and AAC parameters can be either under- or overestimated. Several studies based on simulations of total reflectances were made at the scale of 1 km corresponding to a moderate resolution radiometer such as MODIS or the GLobal Imager (GLI/ADEOS2) to assess errors for liquid water clouds on optical thickness (Iwabuchi and Hayasaka, 2002; Zinner and Mayer, 2006) or on effective radius (Zhang et al., 2012). Kato et al. (2006) analyzed in addition the error on the albedo of the cloudy scenes, which is an important parameter for cloud radiative budget studies. At 1 km pixel size, they found significant errors ranging between -0.3% to 14% (-5% to 30%) from nadir (oblique) viewing depending on the cloud heterogeneity. Some recent studies were also made for ice clouds and found non negligible errors on retrieved COT from InfraRed (IR) measurements (Fauchez et al., 2015) or from visible and near-infrared measurements (Zhou et al., 2017). Concerning aerosol above cloud retrieval, to our known ledge, no study were conducted to assess errors due to cloud heterogeneity.

In this paper, we investigate the impact of cloud heterogeneities on retrieved parameters on observations from the POLarization and Directionality of Earth Reflectance radiometer, POLDER, which was on board the platforms ADEOS1 in 1999, ADEOS2

in 2002 and PARASOL between 2005 and 2013. POLDER/PARASOL allows to measure multi-angular total reflectances from 443 to 1020 nm and multi-angular polarized reflectances for three channels (490, 670 and 865 nm).

A review of the POLDER capabilities for cloud measurements and retrieval are presented in (Parol et al., 2004). Comparisons with MODIS cloud products were analyzed for cloud fraction in Zeng et al. (2011), for cloud phase in Zeng et al. (2013) and cloud optical thickness in Zeng et al. (2012). In the latter, the plane-parallel bias and 3D cloud effects were observed in the COT values retrieved from multi-angle measurements under oblique solar illumination: lower COT were retrieved in the forward viewing direction and larger COT in the backward viewing direction (Figures 8 and 9 in Zeng et al. (2012)). Reflectance simulations from known cloud properties help to understand quantitatively the errors or biases on the retrieved cloud properties. In addition, assessment of POLDER algorithms will be helpful in a near future as the Multi-viewing, Multi-Channel, Multi-Polarization Imaging mission (3MI), a POLDER type follow-on instrument is planned to be part of the future generation of EUMETSAT polar satellites, EPS-SG (Marbach et al., 2015).

Total but also polarized reflectances were simulated at a small scale (50 m) from synthetic 3D cloud fields and averaged at the POLDER pixel size (7 km x 7 km) to simulate POLDER measurements. The different clouds used in our study and presented in Section 2 are generated using an enhanced version of the 3DCLOUD model (Szczap et al., 2014; Alkasem et al., 2017). The POLDER cloud operational algorithm described in (Buriez et al., 1997) is then used to retrieve the COT and the albedo of the cloudy scene. Results are presented in Section 3.

Contrary to MODIS, POLDER does not make measurements in the near infrared to get information on cloud particle size. The two first moments of the cloud droplet distribution are obtained from polarized angular measurements (Bréon and Goloub, 1998; Breon and Doutriaux-Boucher, 2005) as well as the cloud top pressure (Goloub et al., 1994). Polarized reflectance measurements are also used for cloud droplet retrievals by the Research Scanning Polarimeter (Alexandrov et al., 2012). Cloud heterogeneity effects on polarized measurements of liquid clouds have been studied for a single flat cloud in (Cornet et al., 2013) and almost no effects were found. Here, we go further and present in Section 4.1 differences between 3D and 1D polarized angular reflectances for different clouds and geometries. Consequences for 3D cloud radiative effects on the effective radius, effective variance and cloud top pressure retrieval are presented in Section 4.2. The impacts of the 3D effects on the POLDER above cloud AOT operational retrievals in case of fractional cloud were evaluated and presented in Section 4.3. Conclusions are summarized in Section 5.

## 2 Description of the synthetically generated clouds and radiative transfer simulations

The clouds used in this study have been generated with the 3DCLOUD model (Szczap et al., 2014; Alkasem et al., 2017). 3DCLOUD is a fast and flexible algorithm designed for generating realistic 3D extinction or 3D optical thickness for stratocumulus, cumulus and cirrus cloud fields. 3DCLOUD cloud fields share some pertinent statistical properties observed in real clouds such as a gamma distributed optical thickness and the Fourier spectral slope $\beta$ close to $-5/3$ between the smallest scale of the simulation to the outer scale $L_{out}$ where the spectrum becomes flat. In addition, the user can specify the mean

optical thickness COT, the heterogeneity parameter $\rho$ (standard deviation of COT normalized by the mean of COT) and the cloud coverage $C$. In a first step, 3DCLOUD solves drastically simplified basic atmospheric equations and integrates user's prescribed large-scale meteorological profiles (humidity, pressure, temperature and wind speed), in order to simulate 3D cloud structures of liquid water content (LWC). In a second step, the amplitude of the wavelet coefficient of the extinctions are manipulated with a 3D wavelet transform of the whole 3D cloudy volume to constrain the mean COT, $\rho$, $\beta$ and $L_{out}$ (Alkassem et al., 2017).

Here, we generated three cloud fields composed of 140 x 140 pixels with an initial horizontal resolution of 50 m resulting to a 7 km x 7 km field, which corresponds to a POLDER pixel size. The choice of 50m for the pixel scale was made considering the mean free path of the photon, (corresponding to the inverse of the extinction coefficient so to about 70m) but also considering computation time and virtual memory availability.

The three generated clouds have the same mean optical thickness close to 10 at 865 nm. We created two stratocumulus clouds and one cumulus cloud. The latter is the result of instabilities of the boundary layer and lead to fractional cloud cover and larger heterogeneity parameter (Kawai and Teixeira, 2011). The flat and bumpy clouds representing overcast stratocumulus clouds have the same heterogeneity parameter across the 140x140 pixels, $\rho = 0.6$, which is a typical value for stratocumulus cloud. The cumulus cloud has a fractional cloud cover equal to 0.76 and a heterogeneity parameter equal to 1.12 setting clear sky pixels to null values (0.95 if computed only with the cloudy pixels). These values are typical values obtained from Landsat data (Barker et al., 1996) for stratocumulus and cumulus clouds.

Figure 1 shows the vertical profiles of potential temperature and of vapor mixing ratio prescribed in this study to generate the three cloud fields. Globally, the vertical profiles of potential temperature and vapor mixing ratio give the cloud position. The mean cloud top height is mainly determined by the height where the potential temperature increases and the vapor mixing ratio decreases. Cloud top height fluctuations (shapes of top bumps) are mainly the result of the intensity of the vertical gradient of the potential temperature and vapor mixing ratio.

Figure 2 shows the horizontal cloud optical thickness field and a vertical profile through each cloud. In this study, we focus on the effects of the optical thickness heterogeneity, which is supposed, in real cloud, to be more important than the microphysical heterogeneity (Magaritz-Ronen et al., 2016). Consequently, the cloud droplet size distribution is assumed uniform everywhere in the cloud and follows a log-normal distribution with an effective radius of 11 μm and an effective variance of 0.02.

From these 3D cloud fields, we simulated the total and polarized bidirectional reflectances function for the viewing zenith angle $\theta$ and the viewing azimuthal angle $\varphi$. By convenience, in the following, we call them total reflectance $R$ and polarized reflectance $Rp$:

$$R(\theta, \varphi) = \frac{\pi . I(\theta, \varphi)}{F_0 cos\theta_0}$$

$$R_p(\theta, \varphi) = \frac{\pi}{F_0 cos\theta_0} \sqrt{Q^2(\theta, \varphi) + U^2(\theta, \varphi) + V^2(\theta, \varphi)}$$

where $I(\theta, \varphi), Q(\theta, \varphi), U(\theta, \varphi)$ and $V(\theta, \varphi)$ are the four Stokes parameters in W.m$^{-2}$.sr$^{-1}$, $F_0$ the solar flux in W.m$^{-2}$ and $\theta_0$
the solar zenith angle.

Reflectances for three solar incidence angles 20°, 40° and 60° are computed with the 3D radiative transfer model, 3DMCPOL
It is a forward Monte-Carlo model able to compute radiative reflected or transmitted Stokes vector as well as upwelling and
downwelling fluxes in three-dimensional atmospheres. Initially, developed for solar radiation (Cornet et al., 2010), it was next
extended to thermal radiation (Fauchez et al., 2014). To save time and for an accurate computation of reflectances, the local
estimate method (Marshak and Davis, 2005) is used. Periodical boundary conditions at the horizontal domain limits are used.
For highly peaked phase function, the potter truncation is implemented. Molecular scattering is computed according to the
pressure profile. A heterogeneous surface can also be specified with Lambertian reflection, ocean or snow bidirectional
function. The model participated and was improved during the Intercomparison of Polarized Radiative Transfer model (IPRT)
on homogeneous cloud cases (Emde et al., 2015) and on 3D cloud cases (Emde et al., 2018).
Simulations are run with a total of $10^7$ photons and $10^9$ photons for the homogeneous and heterogeneous clouds respectively.
The Monte-Carlo uncertainties are estimated with the computation of standard deviation with 10 and 50 independent
realizations of $10^6$ and $20.10^6$ photons for the homogeneous and heterogeneous cloud respectively. For the homogeneous case,
the relative standard deviation is below 0.12% for the total reflectances and below 1.2% for the polarized reflectances. For the
heterogeneous clouds, at 50m resolution, the mean relative standard deviation is below 1.3% for the total reflectances. For
polarized reflectances at 50 m, the mean relative standard deviation varies according to the angular geometry and is between
2% and 107% for very small reflectance values with a mean value of 23%. At 7km, as the reflectances are averaged, relative
standard deviation values are much lower below 0.01%  and 0.8% for total and polarized reflectances respectively.

At this stage, molecular scattering is integrated but no aerosols. To remain consistent with assumptions made within POLDER
operational algorithm, an oceanic surface with a wind speed of 7 m.s$^{-1}$ is included for total reflectances while a black surface
is included for polarized reflectances. Indeed, for retrieval using polarized reflectances, the multi-angular ability of POLDER
provides the advantage of not using the directions close to the sun-glint where the polarized reflectances can be high..

As POLDER measures up to 16 directions, we simulate reflectances for 16 POLDER typical zenith observation angles in the
solar plane. Total reflectances of the three clouds are presented in Figure 3 (first column) with a 50 m spatial resolution for a

solar incidence angle of 60° in the cloudbow direction (40° from the backward direction). Polarized reflectance fields are discussed in Section 4.1.

## 3 Impacts on total reflectances and consequences for optical thickness and albedo retrieval

We averaged spatially the 50 m resolution reflectances fields at 7 km x 7 km to mimic the radiometer measurements and applied the POLDER operational algorithm on these synthetic measurements to obtain cloud optical thickness and albedo. In order to assess the retrieval errors due to the cloud homogeneous assumption without biases due to differences in reflectance computations, we also computed the 1D reflectances of the three equivalent homogenous clouds, which are subsequently used for retrieval to act as references for the inhomogeneous cloud retrievals. The COT of the equivalent homogeneous clouds is the mean COT of the heterogeneous clouds, and their cloud top and base altitudes correspond to the maximum and minimum altitudes of the respective homogenous clouds.. The mean optical thickness, and the cloud top and base altitudes corresponding to the maximal and minimal altitudes of the heterogeneous clouds are used.

Figure 4 summarizes the results obtained for the retrieved cloud optical thickness for the three solar zenith angles and the four cases, namely the homogeneous (1D), the flat, the bumpy and the fractional cloud. The optical thicknesses are plotted as a function of sensor zenith angle with negative value corresponding to backward scattering directions and positive value to forward scattering directions. The homogeneous cloud values (1D) are only plotted for control and we observe logically that the retrieved value is almost constant and close to 10, independent of the solar incidence angle, since the same assumption (1D homogeneous cloud) is used in both the forward simulation and retrieval algorithm. Slight differences appear because of inclusion of aerosol optical thickness in the forward model used to build the look-up table (Buriez et al., 1997) but not in our simulations. The small angular difference in the backward direction at 20° can be attributed to interpolation in the LUT.

Looking at results concerning the heterogeneous clouds (3D), we clearly note, in the angular range between about -30° and +30°, the plane-parallel bias, which leads to retrieve optical thicknesses lower than the mean optical thickness. At nadir view, the relative error is between -10 and -20% both for the flat and bumpy cloud and is much larger for the fractional cloud, between -35 and -50%. The flat and bumpy clouds were built with the same heterogeneity parameter ($\rho$=0.6) whereas the fractional cloud has a larger heterogeneity parameter including the zeros ($\rho$=1.12) due to its fractional nature. That confirms that heterogeneity parameters can be at first order used to characterize plan-parallel bias (Cahalan et al. 1994, Szczap et al., 2000a).

For solar zenith angle (SZA) equal to 20°, the retrieved optical thickness is almost independent of the observation geometry whatever the cloud type, while for SZA=60°, significant differences between view angles are observed. We note indeed a strong decrease of the retrieved optical thickness value in the forward scattering direction leading to a relative bias on the retrieved optical thickness between -40% for the flat and bumpy cloud and -70% for the fractional cloud. On the contrary, we can notice an increase of the retrieved optical thickness value in the backscatter direction (relative bias ranging from +3% for the flat cloud, +43% for the bumpy cloud and +21% for the fractional cloud). This angular behavior was already simulated by

several authors at the resolution of 1 km (Loeb et al., 1998; Varnai, 2000; Iwabuchi and Hayasaka, 2002; Zinner and Mayer, 2006) and agrees with POLDER observations (Buriez et al., 2001; Zeng et al., 2012). In the backscatter directions, the cloud sides illuminated by the Sun make the cloud brighter, in contrast to the forward direction where cloud sides are in the shadow (Varnai and Davies, 1999). These effects are visible for the bumpy cloud but are much less pronounced for the flat cloud. The heterogeneity parameter thus seems well adapted to characterize quantitatively the plane-parallel bias (Szczap et al., 2000a) but not sufficient to characterize the amplitude of the 3D effects. Indeed, the flat and bumpy clouds, which are characterized by the same heterogeneity parameter value show close plane-parallel bias (about -10-20% for nadir view) but quite different amplitudes of the 3D effects, especially in the backward direction for SZA=60°. We note also that this error in the backward direction is larger for the bumpy cloud (about +40%) compared to the fractional cloud (about +20%) because for the latter the plane-parallel bias is stronger (about -40% at nadir view).

The following step in the POLDER operational algorithm consists in computing the albedo of the cloudy scene, corresponding to the upward flux normalized by the solar incident flux, from the retrieved cloud optical thickness using look-up tables (Buriez et al., 1997). The albedo is not derived from a single view as computed in Kato et al. (2006) at 1 km x 1 km but from all view angles. The multi-angular capabilities of POLDER allow then averaging over the different values using a directional weighting function. The aim of this weighting function is to limit the influence of directions for which the microphysical or 3D effects can be important as for example in the cloudbow, glory and forward directions (Buriez et al., 2005).

The assessment of cloud heterogeneity effects on cloud albedo is realized by comparing the retrieved POLDER algorithm albedos with the ones directly computed with the 3DMCPOL radiative transfer model identified as the true one. Direct comparisons of retrieved albedos values from homogeneous or from the heterogenous clouds as done for other parameters are not suitable for cloud albedo. Indeed, the plane-parallel bias leads to reflectances off of a heterogenous cloud lower than the reflectances off of an equivalent homogenous cloud with the same (mean) COT. The retrieved optical thickness is lower than the mean optical thickness of 10 (Figure 4). Using it to recompute the albedo in the POLDER algorithm leads to a too low value comparing to the albedo of the equivalent homogeneous cloud. Contrarily, using 1D cloud radiative model in the inversion and in the direct computation as it is done in the operational algorithm, is consistent and leads to a sound cloud albedo. The plane-parallel bias is indeed almost cancelled.

Values of the computed and retrieved albedos and their relative differences are indicated in Table 1. The first line (homogeneous cloud) shows very good consistency between the 3DMCPOL radiative transfer code and the retrieved values using the POLDER operational algorithm. Relative differences between computed and retrieved albedos remain smaller than 0.5%.

For SZA= 20°, the POLDER operational algorithm underestimates slightly the albedo for the flat and bumpy cloud with relative differences under -2.5%. The relative error is slightly larger for the fractional cloud (-4.7%). The relative differences are low compared to optical thickness errors because, as explained above, the same cloud model (i.e the homogeneous cloud) is used to retrieve and to compute the albedo. The slight underestimation of the retrieved albedo comes from differences in

the non-linear relationship between reflectances and albedo as a function of the optical thickness. It implies that effects of the plane-parallel bias are not the same for reflectances and albedos. Inversely, for SZA = 60°, the albedo is overestimated by 2.35% for the flat cloud case and 7.88% for the fractional cloud case because illumination effects in the backscattering direction are not completely cancelled by the weighting function.

At SZA=40°, negative differences due to the plane parallel biases are on contrary almost cancelled by illumination effects for bumpy and fractional cloud leading to very small errors of -0.26% and +0.13% respectively.

## 4. Differences between 3D and 1D polarized reflectances and consequences for microphysical distribution, cloud pressure and aerosol above cloud retrievals

### 4.1 Cloud heterogeneity effects on polarized reflectances

As explained before, we simulated using 3DMCPOL, the polarized reflectances for the three wavelengths used in the POLDER retrieval algorithms (e.g. 490, 670 and 865 nm). Total and polarized reflectances at 490 nm for 50 m resolution are presented in Figure 3 (second and third columns) for SZA=60°. First of all, we can see that for flat cloud, the polarized reflectance field appears smoother than the total reflectance field. As polarized reflectances level off for optical thickness greater than about 3, all cloudy pixels with higher optical thickness provide almost the same polarized reflectance. Therefore, cloud heterogeneity

effects are visually less discernible on polarized reflectance fields compared to the total reflectance fields.

For the bumpy or fractional clouds, the polarized reflectance field appears much rougher. In the cloudbow viewing directions (second column), some parts of the cloud facing to Sun appear brighter and other parts in the shadow darker. At this small spatial scale (50 m), a large part of the total amount of pixels exhibits polarized reflectance higher than the maximum value predicted by the 1D homogeneous cloud model (yellow pixels) and thus cannot be obtained with 1D radiative transfer

simulation : at 490 nm, their ratio reaches 41% of the total number of pixels for the flat cloud, 52% for the bumpy cloud and 38% for the fractional cloud. This phenomenon of illumination and shadowing was already highlighted with simply a step cloud in Cornet et al. (2010).

In the forward direction ($\Theta$=60°) at 490 nm (third column in Figure 3), the "shadow areas" are not dark anymore contrarily to the total reflectance images (first column in Figure 3) and appear even brighter than cloudy part. For short wavelength and

25 forward scattering angles, molecular signal is stronger than the cloud signal and thus enhances the polarized signal in the shadow parts.

In Figure 5, we plot the average polarized reflectances as would be measured by POLDER at 7 km x 7 km resolution as a function of the scattering angle $\Theta$ for a solar zenith angle SZA=60°, and for the three wavelengths. As we can see in Figure 5a, the main differences between homogeneous and heterogeneous clouds appear in the cloudbow direction ($\Theta$=140°) and in

the forward direction ($\Theta$ < 80°). In the cloudbow direction, the 3D polarized reflectances are lower than the 1D ones for the three clouds. Similar to the total reflectances, this is mainly due to the plane-parallel bias. In these directions, the relative

differences (Figure 5b) are about -9%, -12% and -35% for the flat, bumpy and fractional cloud, respectively. We note that the relative difference is slightly lower for 490 nm because of the smoothing effects by molecular scattering above the cloud.

In the forward scattering direction, the consequences of the 3D effects in terms of absolute polarized reflectances appear differently depending on the wavelength. At 490 nm, the 3D effects enhance the absolute polarization, while at 865 nm they reduce it. At 490 nm, atmospheric molecular scattering is very strong. The 3D polarized reflectances appear greater than the 1D ones because, as seen in Figure 3, the polarization in the shadow parts of the cloud is enhanced by this molecular scattering. At 865 nm, the shadow parts appear dark with small positive values that reduce the negative polarization of the cloud and consequently the absolute polarization. The relative difference (Figure 5b) is consequently positive for 490 nm (about +55% for the fractional cloud) and negative for 865 nm (about -75% for the fractional cloud). At 670 nm, the polarized reflectance in the shadow part is only slightly enhanced by the molecular scattering but more compared to 865 nm. Polarized reflectances thus become positive for the fractional cloud but not for the flat and bumpy clouds. Note that in the backward direction, the polarized reflectances are very weak so no heterogeneity or 3D effects can be detected.

Figures5 illustrate results obtained for simulations for SZA=60° with a scattering angular range between 60 and 180°. Note that for SZA = 20° and SZA = 40°, the plots are similar with a reduced scattering angular range that is between 100° and 180° for SZA=20° and between 80° and 180° for SZA=40°. Consequently, for SZA = 20 ° and SZA=40 ° the attenuation due to the plane-parallel bias is the main impact of the measurements.

## 4.2 Consequences for droplet size distribution and cloud top pressure retrievals

The polarized signal is used as input of a POLDER retrieval algorithm developed to retrieve effective radius, effective variance and cloud top pressure. It uses the polarized information as presented in Bréon and Goloub (1998). The position of the cloudbow as well as the position of the supernumerary bows gives information on the effective radius. The amplitude of the supernumerary bows gives information on the effective variance of the cloud droplet size distribution. For cloud top pressure, the algorithm uses the information given by the molecular scattering which depends, in the forward scattering directions, on the atmospheric air mass factor (Goloub et al., 1994). The algorithm, under implementation in the POLDER operational algorithm, is based on an optimal estimation method (Rodgers, 2000) and provides errors associated to each of the retrieved parameters. It is also possible to add in the forward model variance-covariance matrix an error due to the non-retrieved parameter. Following previous computations made in (Waquet et al., 2013a) (Waquet et al., 2013a), for the misrepresentation of 3D effects, the error added in the variance-covariance matrix on the reflectances is 7.5% in the directions close to the cloudbow and 5% elsewhere .

The retrieved values obtained with this algorithm based on the homogeneous cloud assumption, are presented in Table 2. We use again the homogeneous cloud (1D cloud) to check the consistency of our simulations. For all clouds, even if differences in polarized reflectances are large in amplitude, the retrieval algorithm still capture the general angular features of the three

wavelengths, which results of small errors on the retrieved effective radius and effective variance. The algorithm is able to retrieve an effective radius of 11 μm and an effective variance of 0.02 with relative error compared to the input under 2.6 % and 2.1% respectively (see Table 2). Indeed, as the cloud heterogeneity effects do not modify the cloudbow position and the number of supernumerary bows, the retrieval of the droplet size distribution parameters is not really affected by 3D effects.

This is a fundamental advantage of the polarized measurements compared to the bi-spectral method (Zhang et al., 2012), usually used when visible and shortwave infrared wavelengths are available. We note, however that the cost function, which is the root mean square difference between the model and measurements weighted by the respective variance-covariance matrix is larger for 3D clouds than for the homogeneous cloud. It means that the forward model (homogeneous model) used for the retrieval does not allow matching perfectly the heterogeneous cloud reflectances used as input. For the bumpy and fractional

cloud, the algorithm does not even converge meaning that the direct model is not able to represent the signal within the allocated uncertainties. The main impact of cloud heterogeneities appears for cloud top pressure retrieval. In table 2, we report the mean cloud top height for each heterogeneous cloud and the retrieved value. The 1D homogeneous values used for control was set the intermediate mean cloud top altitude. We note slight differences about -4 hPa (+ 37m) between input and 1D retrieval, which reveals slight differences between the radiative transfer codes used for the simulation and for the retrieval. Differences

between 3D and 1D are however much larger, especially for the bumpy and fractional cloud with values of +62hPa (-550m) and +45hPa (-390m).

As already explained, the polarized reflectance in the shortwave wavelengths (490 nm) is very high because of molecular scattering. The retrieval of the cloud top pressure is based on the amount of molecular scattering occurring above the cloud when looking in forward scattering (for scattering angle ranging between 60 and 120 degrees). Consequently, as shadowing

effects modify the polarized reflectances in the forward scattering directions, the cloud top pressure retrieval is impacted, especially for the fractional and bumpy cloud. The difference can reach +123 hPa, which means that the cloud seems to be about 1 km lower.

## 4.3 Impacts for aerosol above cloud retrieval

Polarized reflectances of POLDER are also used to retrieve aerosol optical thickness (AOT) of an aerosol layer above cloud

(Waquet et al., 2009, 2013a). (Waquet et al., 2013a) describes two algorithms for Aerosol Above Clouds (AAC) retrieval using POLDER polarization measurements : (i) the research algorithm, that is an optimal estimation method that retrieves a large number of aerosol and cloud parameters, and (ii) the operational algorithm that allows to retrieve the AOT at 865 nm and the Ångström exponent of aerosol above clouds. The "operational algorithm" is the one considered in the present study. This is algorithm is based on LUTs' calculations performed with the successive order of scattering code that assumes a plane-parallel

atmosphere (Lenoble et al., 2007). It uses assumptions on particle microphysics : six fine mode spherical aerosol models (i.e. effective radius varying between 0.09 and 0.24 microns) are considered and a constant complex refractive index of 1.47+0.01i is assumed. The errors due to the assumption made for the complex refractive index are around 20% on average for the AOT (Peers et al., 2015). Maximal relative error may reach 25% in case of extreme aerosol events (AOT > 0.6 at 550 nm). One

additional non-spherical mineral dust model is also considered in the LUTs.

The operational algorithm uses a specific strategy to retrieve aerosol properties above clouds that depends on the aerosol type and also on the available viewing geometries (see figure 4 in Waquet et al., 2013). In case of fine mode particles, the retrieval is restricted to the use of observations acquired for scattering angles smaller than 130° where polarization measurements are highly sensitive to scattering by fine mode particles (such as biomass burning aerosol) and only weakly sensitive to cloud microphysics. In Figure 6, the dashed line show the increase of the polarized reflectances for scattering angles less than 130° when an aerosol layer is present above a cloud. However, non-spherical particles in the coarse mode such as mineral dust particles, cannot be handled with this method as they do not much polarize light. When dust particles are transported above clouds, they reduce the magnitude of the primary cloud bow. The operational algorithm includes thus the primary bow in order to retrieve the above cloud dust AOT. In this case, as the magnitude of the primary cloud bow primarily depends on the cloud droplet effective radius, it must be estimated or included in the retrieval process. Collocated cloud properties from MODIS at high resolution (1 km × 1 km) are used to characterize and to select the cloudy scenes within a POLDER pixel (6 km × 7 km at nadir) and the MODIS cloud products can then be used in the operational algorithm to estimate the droplets effective radius. As the magnitude of the primary cloud bow is only weakly impacted by the choice of the droplet effective variance, this parameter is assumed to be constant and equal to 0.06. Several filters are eventually applied to obtain a quality-assessed product. For instance, the retrievals are restricted to cloudy pixels associated with cloud optical thicknesses larger than 3.0, since the polarized radiation reflected by the cloud layer is then saturated and does not depend anymore on the cloud optical thickness. Criteria are also used to reject inhomogeneous and fractional cloudy pixels and to avoid cirrus cloud contamination. We refer to Sect. 3.4 in Waquet et al. (2013) for a detailed description of the operational algorithm.

In the POLDER operational algorithm, the underneath cloud is assumed to be homogeneous. Empirical criterions are used to reject heterogeneous and fractional cloudy pixels but a misclassification of the cloudy scenes is still possible. Moreover, it is also important to evaluate the AOT retrieval errors due to 3D effects in case of fractional cloud covers. These scenes, for which aerosols and clouds are potentially mixed, remain untreated and are of primarily importance for climate studies. In the following, we investigate the possibility to use the operational algorithm to treat these scenes and we evaluate the biases observed in the polarized reflectances and in the AOT retrieval errors due to 3D effects. In order to check the AOT value retrieved for such cases, we use the 3D polarized reflectances generated for the fractional cloud case, with and without aerosol, and we used these 3D simulations as inputs for the operational algorithm. Note, that for the synthetic retrievals discussed here below, we assumed that the operational algorithm knows the effective radius and effective variance of the cloud droplets.

The 3D polarized reflectances used as input of the algorithm and the ones simulated after the adjustment of the aerosol model and optical thickness are plotted in Figure 7 (solid lines). When a large scattering angular range is available (between 60 and 180°), the algorithm works in an efficient way The lateral polarized reflectances in scattering angular range between 80° and 120° exhibit low or negative values. Consequently no aerosol (AOT=0) were retrieved. We note however that the primary

cloudbow is not well modelled by the 1D simulation provided by the operational algorithm. In the POLDER measurements, the range of sampled scattering angles varies with the geographical position. In some cases, the scattering angle range sampled by the instrument can be quite narrow. We tested the algorithm without observations acquired for scattering angles smaller than 120° (dashed lines in Figure 7). The cloudbow signal is then better matched but the inversion method retrieves erroneous AOT values of 0.31 at 670 nm and 0.28 at 865 nm instead of zero for both.

A second test is made with simulated reflectances including a biomass-burning aerosol layer lofted above the fractional cloud. For the simulation, the AOT of the aerosol layer is fixed to 0.28 and 0.15, the single scattering albedo to 0.93 and 0.91 at 670 and 865 nm respectively. In order to avoid retrieval errors related to the choice of aerosol model, we used one of the biomass burning aerosol model included in the fast algorithm. The particles effective radius is 0.15 microns and the single scattering albedo is equal 0.91 at 865 nm. The simulated 3D angular polarized reflectances as a function of the scattering angles are presented in Figure 6 (solid blue and red lines). Compared to the 1D reflectances with aerosols above cloud (dashed blue and red lines), the cloud heterogeneity effects amplify the increase of the forward signal and the decrease of the cloudbow signal. As with molecular scattering (Section 4.1), aerosol scattering contributes to enhance the polarized reflectances in the shadow and cloud-free parts leading to higher averaged polarized reflectances in the forward direction In the cloudbow direction (near 140°), and, to a lesser extent, in the side scattering (between 100° and 130° in scattering angle), the polarized reflectances are additionally attenuated because of the plane-parallel biases. Note that for other solar zenith angles, the plots are similar with a more restricted scattering angular range (between 100° and 180° for SZA=20° and between 80° and 180° for SZA=40°). Consequently, only the attenuation due to the plane-parallel bias impacts the measurements.

The results obtained with the operational algorithm are presented in Table 3. We remind that the same input AOT is used in the 1D and 3D simulations (AOT of 0.15 at 865 nm). As expected, the AOTs retrieved by the algorithm for homogenous clouds (1D input) are close to the input one, whatever the SZA value. The retrieved AOTs only slightly overestimate the input one (0.15) and are respectively equal to 0.18, 0.17, 0.17 for SZA of 20, 40 and 60°. This overestimation is likely due to the approximations used in the retrieval algorithm (e.g. interpolation of the LUTs). Comparing with the retrieved values from homogeneous cloud, significant departures are observed for fractional clouds (3D input) depending on the SZA. The AOTs retrieved at 865 nm are then equal to 0.119, 0.17 and 0.28 for SZA of 20, 40 and 60°, respectively. For a given solar zenith angle, the viewing geometries and the angular resolution are identical for the 1D and 3D. The differences observed in AOT between the 1D and 3D calculations are then necessarily due to 3D effects.

The difference of AOT retrieval between 1D and 3D inputs depends on the solar zenith angle. Note that in the Table 3, the Ångström exponent is related to the ratio of two optical thicknesses at two wavelengths and corresponds in the retrieval to the best-selected model.

For SZA=40°, the best model that minimized the cost function is the same for the homogeneous and fractional cloud. Differences for the retrieved AOT are negligible, but we note that the RMSE between the input and recalculated reflectances is slightly larger for the fractional cloud than for the homogenous one.

For SZA = 20°, the operational algorithm also successfully retrieves the input aerosol model for the homogeneous and fractional cloud. However, the AOT retrieved by the operational algorithm, under the 1D assumption, is underestimated with error between -35 and -40%. For a SZA of 20°, the range of scattering angles effectively used for the retrieval is between 100° and 130°. Polarized reflectances for SZA=20° are not shown but they are similar to the ones shown in Figure 7 between 100 and 180°. Over the 100-130°, as shown in Figure 7, 3D polarized reflectances are lower than the 1D ones because of the plane-parallel biases, which explains why the AOT retrieved by the algorithm is underestimated. However, as the differences are mainly due the plane-parallel bias, which is similar for the two wavelengths, the cloud heterogeneity effects do not affect the selection of the best aerosol model.

For SZA = 60°, the range of scattering angles used is between 60° and 130°. Between 60° and 90°, there is an increase of the forward scattering signal due to 3D effects, which is interpreted by the operational algorithm as an increase in the AOT. We note also that 3D effects bias the aerosol model for this case as a smaller value of Ångström exponent (corresponding to a larger effective radius) is retrieved for the fractional cloud. The retrieved AOT is thus higher (AOT of 0.28 comparing to 0.17) with a relative error up to 65%. For SZA=60°, the 3D effects consist in an increase of the polarized signal because of additional scattering in the clear sky parts. This increase is higher at 865 nm than at 670 nm. This leads to the selection by the algorithm of an erroneous model with a smaller Angström exponent.

Note that, in the operational algorithm, the algorithm is not applied for pixels too heterogeneous. Those are filtered using the standard deviation of the COT retrieved at 1 km by MODIS that should not exceed 5. For the fractional cloud of this study, we checked the standard deviation value computed from the input cloud optical thickness (different from the retrieved one) and found 7. It is slightly above the homogeneity limit fixed in the aerosol above cloud algorithm developed for POLDER (Waquet et al., 2013a). The results presented here for aerosol above cloud retrieval can thus be seen as an upper limit for the operational algorithm.

## 5. Conclusion

This study used simulations to understand and quantify effects of cloud heterogeneities on POLDER total and polarized reflectances. We investigate the consequences of heterogeneous cloud radiative effects on the retrieved values of cloud optical thickness, droplet effective radius, effective variance, cloud pressure and optical properties (optical thickness and Angstrom coefficient) of above cloud aerosol, provided by operational and research algorithms of the POLarization and Directionaly of Earth Reflectance (POLDER) instrument. 3D cloud fields were generated with the 3DCLOUD model (Szczap et al., 2014) and the 1D and 3D radiative transfer simulations were done with the Monte Carlo 3DMCPOL model (Cornet et al., 2010). Three types of heterogeneous water cloud were studied: a flat, a bumpy and a fractional cloud.

The reflectances simulated at small spatial scale (50 m) and averaged at the POLDER spatial scale (7 km x 7 km) are used as realistic input of the different cloud operational and research algorithms. For high solar illumination (SZA=20°), the optical thickness retrieval yields, as it was already shown in numerous studies, lower optical thickness than the averaged ones because

of the plane-parallel bias. For POLDER, the retrieved optical thicknesses are underestimated by 10 or 35% depending on the cloud type. For oblique solar incidence, the POLDER algorithm yields higher optical thickness in the backscattering direction due to solar illumination effects and much lower optical thickness (up to -70% for the fractional cloud) in the forward scattering direction due to shadowing effects. The errors on albedo are weaker with largest bias for albedo between -5% for high solar illumination and +8% for solar zenith angle of 60°.

We next analyzed, for the first time, the cloud heterogeneity effects on polarized reflectances. We showed a reduction of the cloudbow and side reflectances due to the plane-parallel bias and the shadowing effects. In the forward scattering direction, the effects are spectrally dependent. For the shortest wavelength (490 nm), the molecular scattering in the shadow areas increases the averaged polarized signal and leads to an increase of the polarized reflectances. At 865 nm, the weak positive polarized reflectances of the shadow areas reduce the polarization of the clouds, which is negative for these scattering angles. However, even if the polarized angular signature is modified, the retrieved effective radius and effective variance are hardly affected because cloud heterogeneities do not modify the positions of the cloudbow and supernumerary bows. The Rayleigh cloud top pressure is, in contrast, biased for a solar zenith angle of 60° by about 120 hPa corresponding to a cloud 1 km lower in the atmosphere.

We also tested the aerosol above cloud algorithm (Waquet et al., 2013a). Even in the absence of aerosol, the algorithm retrieves non-negligible AOT values when only larger scattering angles (between 120 and 180°) are available. With aerosols above a fractional cloud, the AOT can be underestimated for a high solar elevation (SZA=20°) because of the plane-parallel bias and on contrary overestimated for low solar elevation (SZA=60°) because of the shadowed effects that increase polarized reflectances. The Angström exponent is affected by these shadowing effects for SZA=60° but not by the plane-parallel bias since the plan-parallel biases for 490 nm and 865 nm is almost spectrally neutral and since the information used to select the aerosol model is related to the ratio of two wavelengths.

These results mainly show that 3D effects for fractional clouds are primarily significant at forward scattering geometries in case of low solar elevation (scattering angle < 80° and SZA of 60°) and in the rainbow region (scattering angle of about 140° +/- 5°). The range of scattering angles sampled between 60 and 80° is not necessarily useful for an accurate retrieval of the above cloud AOT. So, reducing the range of scattering angles to scattering angle values larger than 80° will help to reduce the errors associated with the AOT retrievals. The algorithm largely overestimates the AOT when the primary bow is included in the retrieval process and when forward and side scattering viewing geometries are not available. This result suggests that polarized measurements acquired for this configuration should not be used for AAC properties retrievals, at least with a retrieval algorithm based on 1D calculations.

Assessment of retrieval errors due to cloud heterogeneity is challenging for the next generation of retrieval algorithms. Indeed, in the future, it appears crucial to have not only values of retrieved parameters but also estimations of their uncertainties. Realistic simulations with known input parameters are very useful tools to assess accurately theses errors including their

dependence on the available angular sampling. Such simulations can also be used to test the next generation of operational algorithms.

Further that assessments of cloud heterogeneity uncertainties, more complex methods should also be developed to retrieve aerosol and cloud properties accounting for the cloud heterogeneities. Several theoretical or case studies have already been conducted. Some tends to mitigate cloud contamination for aerosol property retrieval (Davis et al., 2013; Stap et al., 2016b). Others aim to use 3D radiative transfer model to retrieve 3D cloud properties and hence account for some cloud heterogeneity effects. It requires then more complex inversion methods. Feasibility studies has been conducted using neural network method (Cornet et al., 2004, 2005), 3D tomography with a surrogate function (Levis et al., 2015, Levis et al. 2017) or adjoint method (Martin et al., 2014; Martin and Hasekamp, 2018). The latter two methods are very promising but have been developed in the framework of high resolution measurements (ten to hundred meters) involving no or small plane-parallel bias. They are so not directly applicable to POLDER/PARASOL measurements.

The Multi-viewing, Multi-Channel, Multi-Polarization Imaging mission (3MI) that will fly on METOP-A SG as part of EUMETSAT Polar System after 2021, will have a spatial resolution of 4 x 4 km. The plane-parallel bias is thus expected slightly lower than for the POLDER instrument. In addition, as 3MI will be on the same platform as the Visible Infrared Imager (VII), a multispectral radiometer with a resolution of 500 m, the correction of the plane parallel biases may be possible while the multi-angular capability of 3MI would help to detect the illumination and shadowing effects.

## 6. Acknowledgements

This work has been supported by the French Programme National de Télédétection Spatiale (PNTS, http://www.insu.cnrs.fr/pnts) grant N° PNTS-2014-02 and by the Centre National d'Etudes Spatiales (CNES).

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

| Albedo of the cloudy scene | Sun incidence | SZA=20° | SZA=40° | SZA=60° |
|---|---|---|---|---|
| Homo Cloud (1D) | Simulation | 0.434 | 0.498 | 0.601 |
| | Retrieval | 0.434 | 0.496 | 0.600 |
| | Error (%) | **-0.04** | **-0.46** | **-0.16** |
| Flat Cloud | Simulation | 0.390 | 0.458 | 0.556 |
| | Retrieval | 0.382 | 0.445 | 0.569 |
| | Error (%) | **-2.09** | **-2.80** | **+2.35** |
| Bumpy Cloud | Simulation | 0390 | 0.451 | 0.562 |
| | Retrieval | 0.380 | 0.450 | 0.583 |
| | Error (%) | **-2.44** | **-0.26** | **+3.69** |
| Fractional Cloud | Simulation | 0.301 | 0.353 | 0.475 |
| | Retrieval | 0.287 | 0.353 | 0.513 |
| | Error (%) | **-4.71** | **+0.14** | **+7.88** |

5 **Table 1: For each cloud case, albedo of the cloudy scene obtained from simulation with 3DMCPOL (first line), retrieved with the POLDER operational algorithm (second line) and relative differences [(Retrieval-Simulation)/Simulation x 100] between the two values (third line) for the homogeneous cloud (for control), for the flat, bumpy and fractional clouds for three solar zenith angles (20, 40 and 60°). The mean optical thickness of each cloud is 10 and the effective radius**

| | Input | Homogeneous cloud (1D) | Flat cloud | Bumpy cloud | Fractional cloud |
|---|---|---|---|---|---|
| Reff (µm) | 11.00 | 11.04 | 11.12 | 11.08 | 11.33 |
| Veff | 0.020 | 0.020 | 0.021 | 0.019 | 0.023 |
| Mean CTOP (hPa) /(km) | 873/1.19 | | 903/0.92 | | |
| | 863/1.28 | 859/1.32 | | 925/0.73 | |
| | 901/0.94 | | | | 946/0.55 |
| Cost function | | 8.45 | 30.07 | 63.43 (NC) | 351.4 (NC) |

**Table 2: Retrieved cloud droplet effective radius (Reff), effective variance (Veff) and cloud top altitude (CTOP) from polarized reflectances with an optimal estimation algorithm. First column is the input, second column the retrieval for the homogeneous cloud (1D), third column for the flat cloud, fourth column for the bumpy cloud and fifth column for the fractional cloud. Last line is the final cost function with NC meaning no convergence. The solar zenith angle is 60°. Note that the cloud top altitude is different according to the heterogeneous cloud leading to three different lines.**

|  |  | SZA=20° | SZA=40° | SZA=60° |
|---|---|---|---|---|
|  | Sun incidence | SZA=20° | SZA=40° | SZA=60° |
| AOT670 | Homogeneous cloud | 0.337 | 0.319 | 0.319 |
|  | Fractional cloud | 0.225 | 0.319 | 0.491 |
|  | Difference (%) | **-33.2** | **0.00** | **+53.9** |
| AOT865 | Homogeneous cloud | 0.180 | 0.170 | 0.170 |
|  | Fractional cloud | 0.119 | 0.170 | 0.280 |
|  | Difference (%) | **-33.9** | **0.00** | **+64.7** |
| Angström coefficient | Homogeneous cloud | 2.46 | 2.46 | 2.46 |
|  | Fractional cloud | 2.46 | 2.46 | 2.20 |
|  | Difference (%) | **0.00** | **0.00** | **-10.6** |
| RMSE | Homogeneous cloud | 0.0056 | 0.0043 | 0.0031 |
|  | Fractional cloud | 0.0091 | 0.0053 | 0.0037 |

**Table 3: Retrieved aerosol properties for a biomass aerosol layer above the fractional cloud with the operational algorithm described in (Waquet et al., 2013a) : aerosol optical thickness at 670 nm (AOT670), at 865 nm (AOT865) and Angström coefficient for three solar zenith angles (SZA). Last two lines, RMSE computed between the input and recalculated polarized reflectances for the homogenous and fractional cloud.**

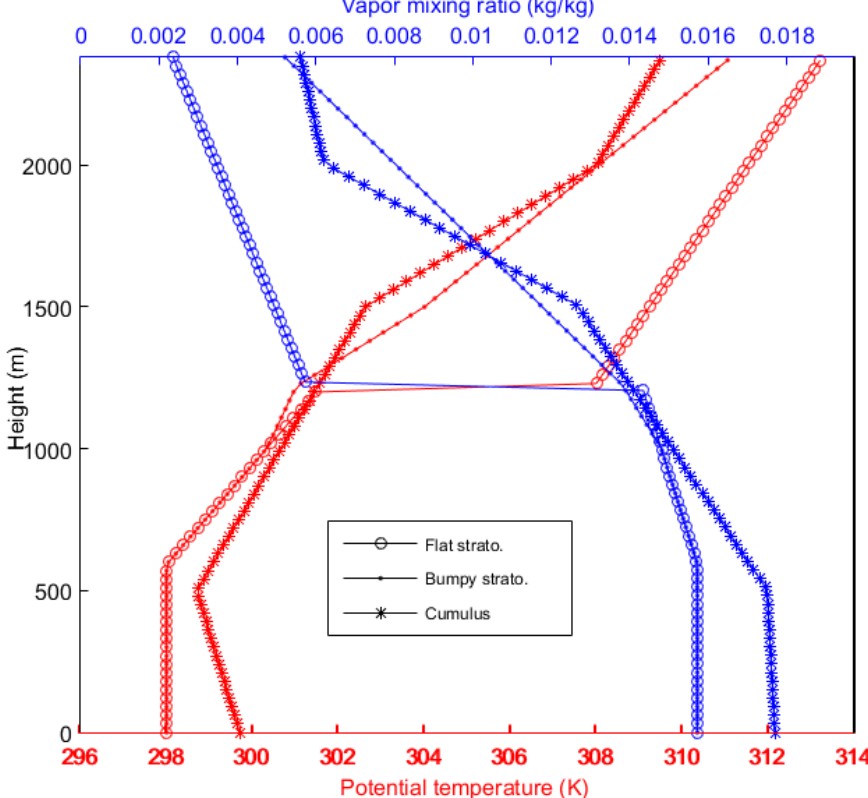

**Figure 1: Vertical profiles of potential temperature and of vapor mixing ratio prescribed in this study to generate the flat stratocumulus (circle), the bumpy stratocumulus(point) and the cumulus (star) cloud fieldsmeteorological profiles to generate to the three cloud fields.**

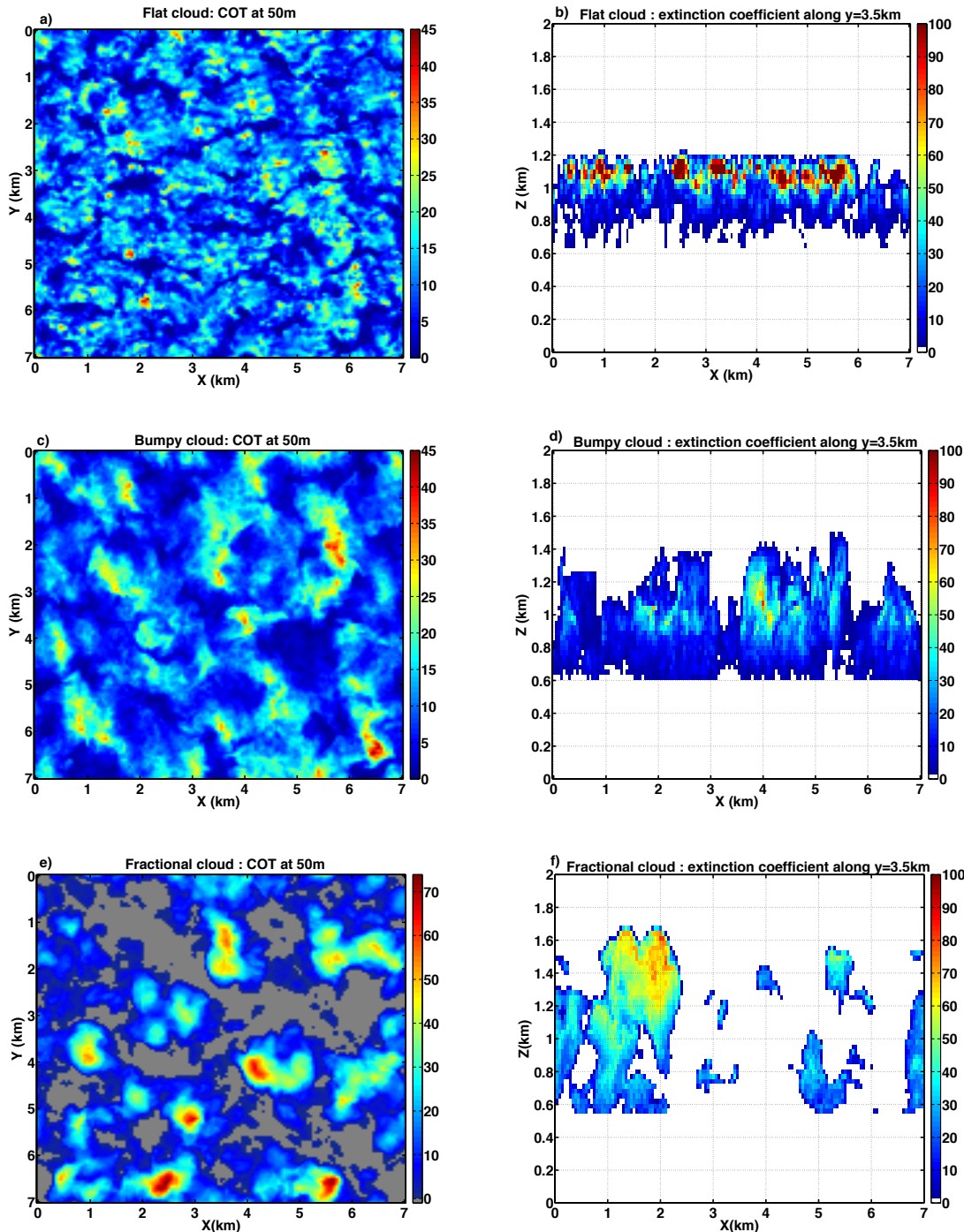

**Figure 2: Cloud optical thickness (COT) of the three clouds used for the study (a) the flat cloud, (c) the bumpy cloud and (e) the fractional cloud. Extinction coefficient (km⁻¹) along the x-z axis for y=3.5 km for the flat cloud (b) the bumpy cloud (d) and the fractional cloud (f).**

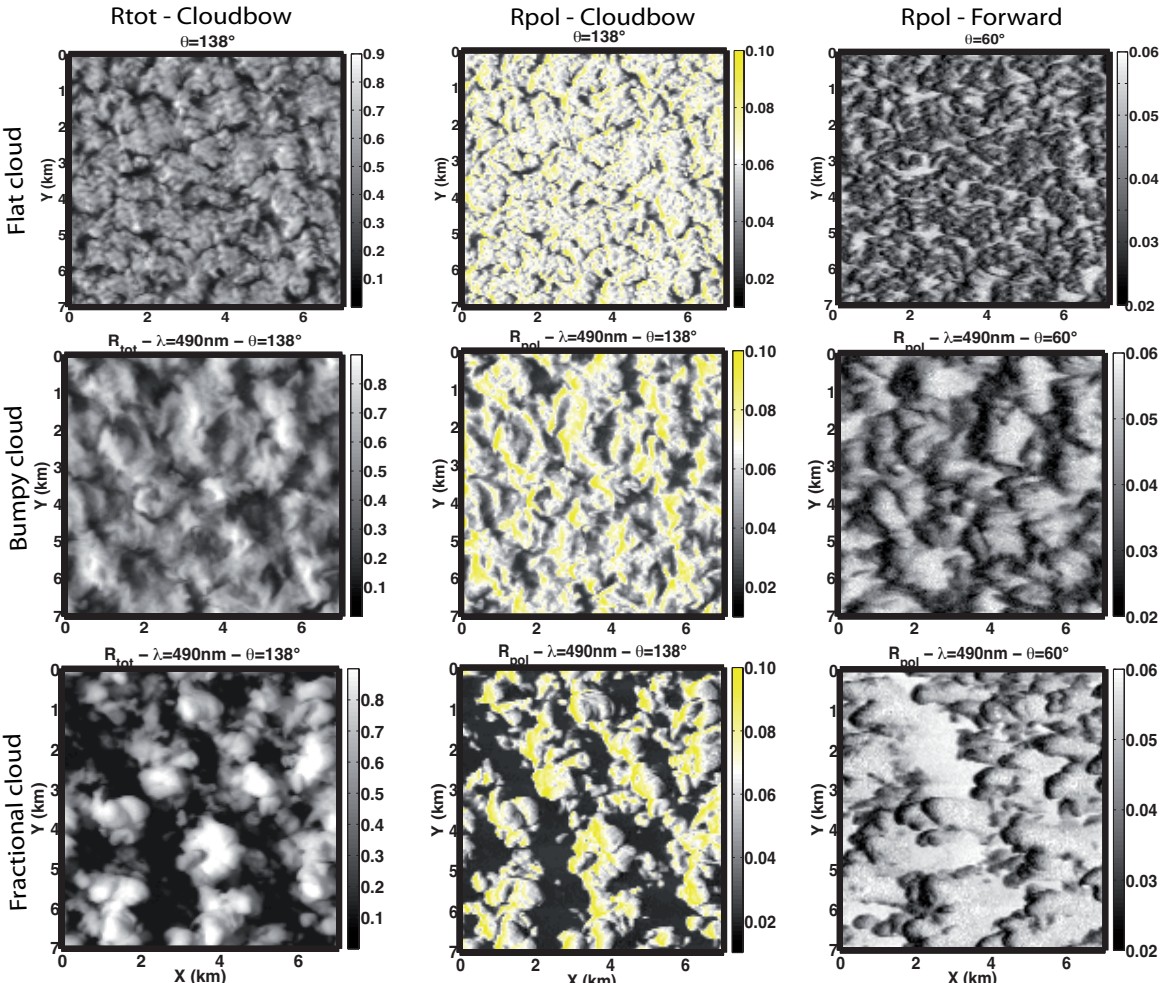

**Figure 3: Total and polarized reflectances for the flat cloud (first line), the bumpy cloud (second line) and the fractional cloud (third line). Total reflectances at 490 nm in the cloudbow scattering direction (first column), polarized reflectances at 490 nm in the cloudbow direction (second column) and polarized reflectances at 490 nm in the forward direction (third column). The Sun illuminates the scene from the left of the Figures (SZA=60°). For polarized**
10 **reflectances in the second column. Yellow color corresponds to polarized reflectance values higher than the maximum value predicted with the homogeneous cloud assumption.**

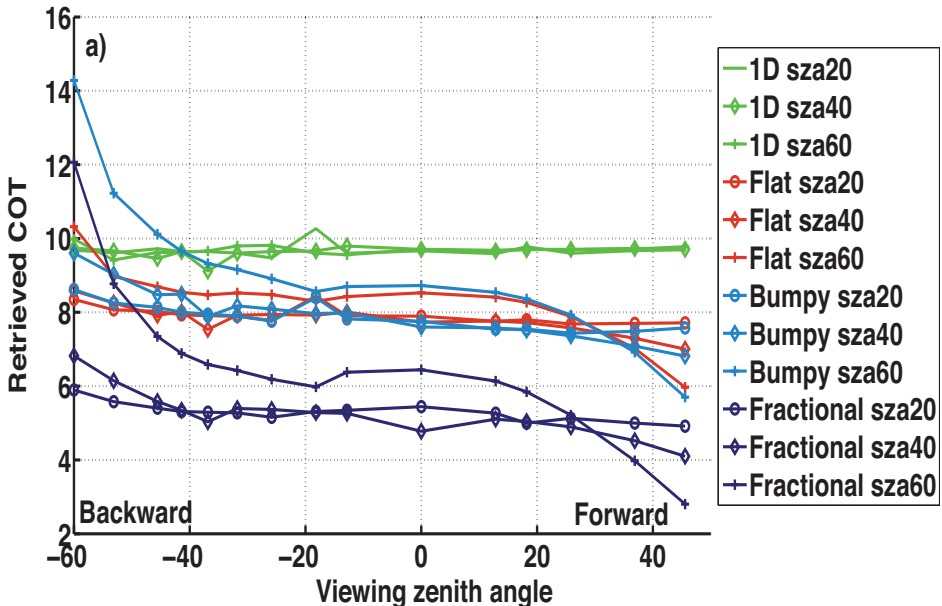

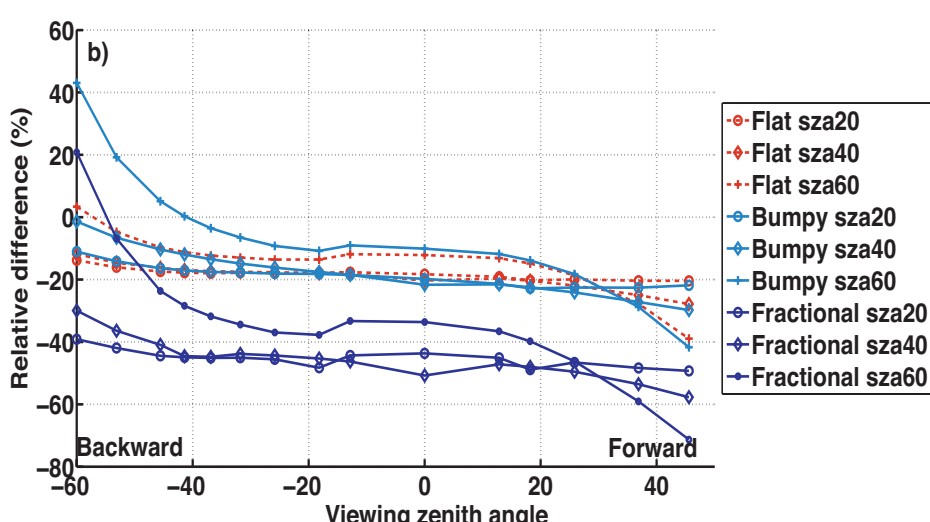

**Figure 4 (a): Cloud optical thickness (COT) retrieved with the POLDER operational algorithm as function of the viewing zenith angle for the four different simulated cloud cases (1D, flat, bumpy and fractional clouds) and for different solar zenith angles (20, 40 and 60°). (b) Relative differences [(COT3D-COT1D)/COT1D x 100] between the heterogeneous cloud (3D) and the homogenous cloud (1D) COT.**

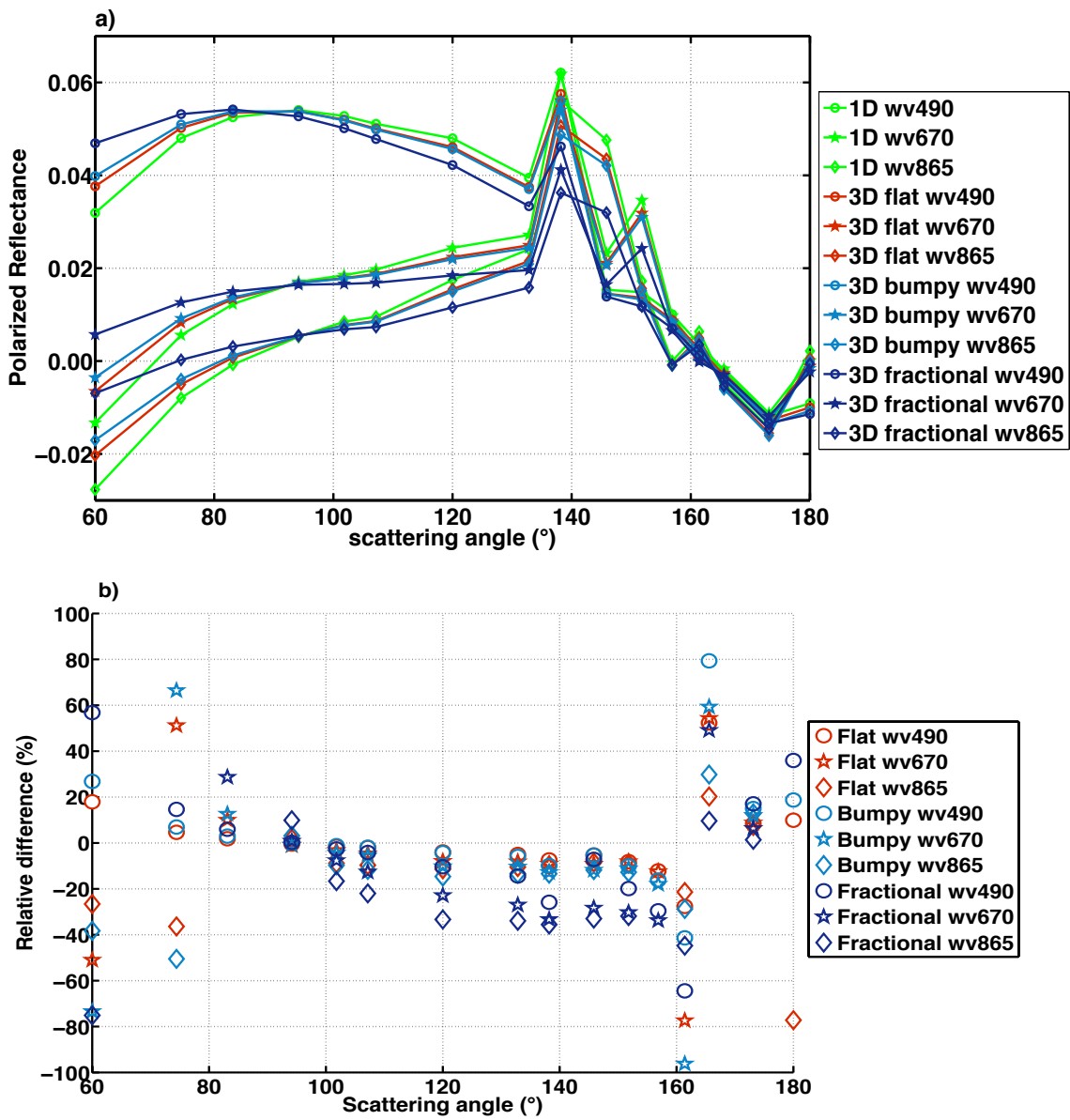

**Figure 5: Polarized reflectance as a function of the scattering angle for three wavelengths (490 nm, 670 nm, 865 nm) for the homogeneous cloud (1D), the flat cloud, the bumpy cloud and the fractional cloud (a). Relative difference between 3D and 1D polarized reflectances, (Rp3D-Rp1D)/Rp1D*100 (b). The solar zenith angle is 60°.**

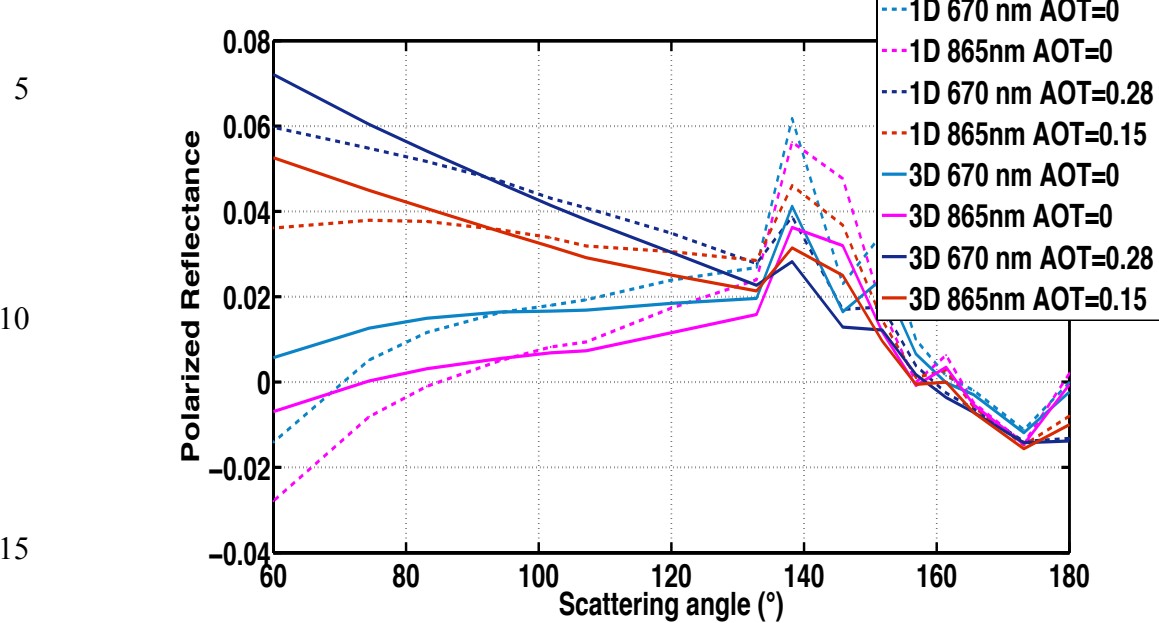

**Figure 6: Polarized reflectances as a function of the scattering angle. Dashed lines are for homogeneous cloud without and with a biomass burning aerosol layer above; solid lines are for the fractional cloud without and with a biomass burning aerosol layer above. The solar zenith angle is 60°.**

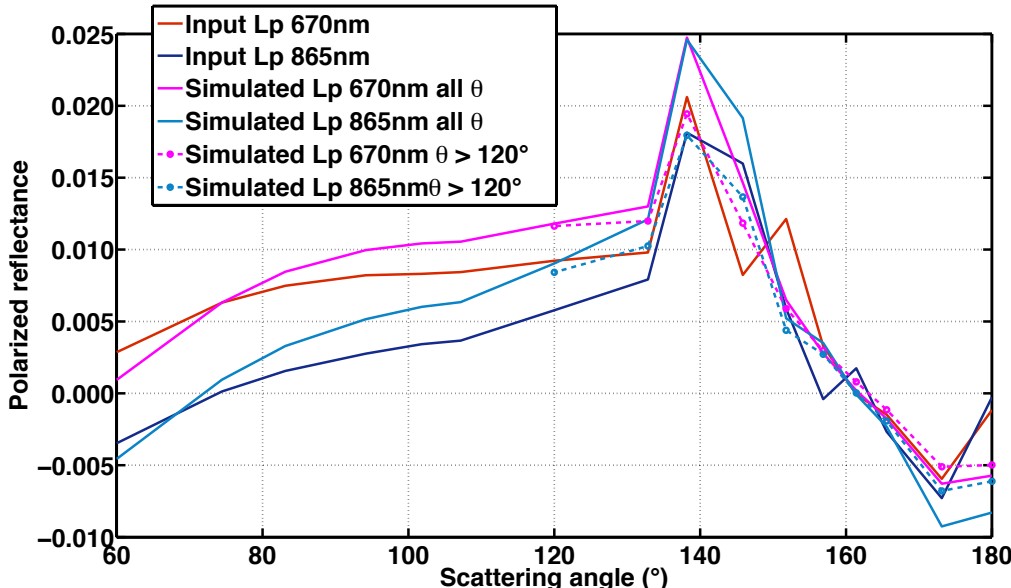

**Figure 7: 3D Polarized reflectances used as input for the Aerosol Above Cloud algorithm (Waquet et al., 2013a) and polarized reflectances simulated with the algorithm after the convergence of the retrieval. Reflectances at all angles were used (solid line) and reflectances with only scattering angles above 120° (dotted line).**

