# Peer review of "Cloud heterogeneity on cloud and aerosol above cloud properties retrieved from simulated total and polarized reflectances"

_Atmospheric Measurement Techniques, 2017_

## Referee Comment (RC1) · G. van Harten (Referee) · 27 Jan 2018

============== General comments:

The goal of the study is to quantify the impact of subpixel cloud heterogeneity on retrievals of various cloud and aerosol parameters. Four clouds with different heterogeneities are compared: a homogeneous, a flat, a bumpy, and a fractional cloud. The procedure is clear:

- The clouds are simulated using the 3DCLOUD cloud model at a 50x50-m resolution.

- Their reflectances are computed using the 3DMCPOL radiative transfer model.

[Figure]

- These reflectances are averaged to 7x7-km resolution to mimic POLDER observations.

- The POLDER operational algorithm is applied to retrieve cloud and aerosol parameters.

- The retrieval results are compared to the truth.

- Interpretation of intermediate results is provided, mostly by comparison to the homogenous cloud.

This work is very relevant for the interpretation of POLDER and future 3MI data, as well as for the optimization of the retrieval algorithm to minimize retrieval bias, e.g. by weighting angular information. The paper is well written, the work is put into context, and the various intermediate results and interpretation thereof ensure good understanding for the reader.

=============== Specific comments:

To draw the right conclusions on the effects of heterogeneity, it is important that the clouds are as similar as possible, except for their heterogeneity. Listed below are comments related to the choice of simulation parameters:

Page 4, line 23-24: Why are the rho's for the fractional cloud not closer to 0.6 for better comparison to the flat and bumpy cloud?

Page 4, line 28: How are the flat and bumpy clouds parameterized? What are the settings for cloud top height, etc?

Page 5, line 6: Why a black surface for polarized reflectances? The surface seems important in particular for the fractional cloud. At certain angles it can be very bright in polarization (sun glint).

Page 5, line 15; Page 18, CTOP: Max(z) / min(z) for cloud top height / bottom of the 3D clouds does not seem like a representative value to me. See Fig. 1: realistic values are

closer to 1.2 (Fig. 1 only shows y=3.5 whereas the realistic value should be computed from all y). Better values should be used, or at least the retrieval results should be compared to more than just max(z).

Page 11, line 3-5: This belongs in Section 2 to put the synthetic clouds into perspective. Apparently, the fractional cloud with stdev(COT)=7 exceeds POLDER's homogeneity limit of 5. The fractional cloud also gives the worst results compared to the flat and bumpy clouds. I think it would be good if the choice for stdev(COT)=7 would be justified in Section 2, and if at least rough numbers are given for how the results compare to a similar fractional cloud with stdev(COT)=5.

============== Detailed comments:

============== Title: "Assessment of A on B" is not correct. Suggestion: "Effects of Cloud Heterogeneity on the Multi-angular Total and Polarized Reflectances from POLDER3/PARASOL, and impact on retrievals of cloud and aerosol above cloud parameters"

============== Abstract: 18: remove "the well-known"

27: remove "well-known"

31-32: "Above the . . . can be underestimated . . . plane-parallel bias." Specify angles. Be more specific than "can be", because that sounds like "can also not be".

============== Page 2:

13: "implemented or under implementation" Which ones are implemented and which are under implementation?

19: "conducts to" -> "leads to"

27: "AVHRR" spell out first time

27: "solar zenith angle elevation" -> "solar zenith angle"

[Figure]

28: "disseminating through the clouds along the cloud side (or bump), and leads to" -> "received along the cloud side (or bump) and disseminated through the cloud, leading to"

30: "which depends on" -> "which depend on"

============== Page 3:

4: "under and/or overestimated" -> "under- or overestimated"

9: "from nadir (oblique view)" -> "for nadir (oblique) viewing"

10: "made in case of ice" -> "made for ice"

10: "IR" spell out

11: "or or from" -> "or from"

16: "1020nm" -> "1020 nm" Fix this throughout paper.

20: "solar illumination, lower" -> "solar illumination: lower"

22: "Reflectances simulations" -> "Reflectance simulations"

26: "Accordingly, total but also polarized" -> "Total and polarized"

28: "in section 2" -> "in Section 2" Capitalize Section, Table, Figure throughout paper.

31: "Contrarily to" -> "Contrary to"

34: "reflectances measurements" -> "reflectance measurements"

34: "droplets retrievals" -> "droplet retrievals"

============== Page 4:

1: "have, been" -> "have been"

3: "Consequences . . . on" -> "Consequences . . . for"

4: "variance cloud" -> "variance, and cloud"

6: "also studied presented" -> "also presented"

7: "synthetic generated" -> "synthetically generated"

13: "optical depth COT" -> "optical thickness COT"

13: "inhomogeneity parameter" -> "heterogeneity parameter" (this word is used later on)

17: "to constrain the intensity of the mean" -> "to constrain the mean"

19: "clouds fields" -> "cloud fields"

21: "heterogeneity parameter … of 0.6" -> "heterogeneity parameter across the 140x140 pixels of rho=0.6"

23-24: Why are the rho's for the fractional cloud not closer to 0.6 for better comparison to the flat and bumpy cloud?

26: "which is supposed, in real cloud, to be more important than" Reference? How much more important? Which retrieval parameters are affected?

27: "cloud size" -> "cloud droplet size"

28: How are the flat and bumpy clouds parameterized? What are the settings for cloud top height, etc?

============== Page 5:

1. "I(theta,phi) are" -> "I(theta,phi) is"

1. steradian "sr" with small "s"

5. "7 m.s-1" "-1" superscript

6. Why a black surface for polarized reflectances? The surface seems important in

particular for the fractional cloud. At certain angles it can be very bright in polarization (sun glint).

8: "Figure 3" -> "Figure 2" I propose to swap Figures 2 and 3, because currently Fig. 3 is mentioned before Fig. 2. Also swap all the references to Figs. 2 and 3.

9: "Polarized reflectances fields" -> "Polarized reflectance fields"

11: "reflectances fields" -> "reflectance fields"

14: "the equivalent homogenous cloud. " -> "three equivalent homogenous clouds, which are subsequently used for retrieval to act as references for the inhomogenous cloud retrievals. The COT of the equivalent homogenous clouds is the mean COT of the heterogenous clouds, and their cloud top and base altitudes correspond to the maximum and minimum altitude of the respective homogenous clouds."

15: Max(z) / min(z) for cloud top height / bottom of the 3D clouds does not seem like a representative value to me. See Fig. 1: realistic values are closer to 1.2 (Fig. 1 only shows y=3.5 whereas the realistic value should be computed from all y). Better values should be used, or at least the retrieval results should be compared to more than just max(z).

17: "e.g. the" -> "namely the"

18: "algorithm retrieves" Does it really retrieve cloud cover, or should it say "algorithm assumes"?

21: "independently of" -> "independent of"

21: "assumption (e.g. 1D" -> "assumption (1D"

22: "the direct simulation" -> "the forward simulation"

23-24: "but not into our" -> "but not in our"

26: remove "well-known". Rather spend one sentence explaining it.
============== Page 6:

4: "differences are important according to the view direction" -> "significant differences between view angles are observed"

10: "pretty well" . . .

11: "the sun making the" -> "the Sun make the" Capitalize Sun consistently.

11: "brighter than in" -> "brighter, in contrast to"

11-12: remove "on the contrary"

13: "seems thus" -> "thus seems"

22-23: "from every view" -> "from all view"

28: "optical thickness lower than" -> "optical thickness, lower than"

33: "cloud fields" -> "cloud field"

============== Page 7:

4: "the one used for the" -> "the retrieved values using the"

7: "the relative is" -> "the relative error is"

8: "quite low" -> "low"

10: delete "(curvature degree)"

10: "in function of" -> "as a function of"

11: "It involves" -> "It implies"

11: "bias is not" -> "bias are not"

12: "2.35 %" -> "2.35%"

15: I found the paragraph about albedo hard to read. If I understand correctly, the train

of reasoning is: - In order to retrieve cloud albedo, the POLDER retrieval algorithm first retrieves COT from multi-angle reflectances, then does a forward computation from COT to albedo. - A heterogeneous cloud has lower reflectance than a homogenous cloud with the same (mean) cloud optical depth, due to plane-parallel bias. - The POLDER algorithm will thus retrieve a COT that is too low. - From that too-low COT, the POLDER forward computation will produce an albedo that is also too low if the cloud were really a homogeneous cloud, but since it is really a heterogeneous cloud with a lower albedo due to plane-parallel bias, the POLDER-retrieved albedo is actually very close. - To compare this retrieval result to the "truth", the actual albedos are directly calculated from the 3DMCPOL radiances. It would be helpful if this could be explained in a more direct way, e.g. by preparing the reader by summarizing this at the beginning of the paragraph, before going into the details. Other suggestions for textual changes:

- Page 6, line 26: "3D reflectances and from 1D reflectances are not comparable" -> "a heterogeneous cloud are not the same as the ones retrieved from an equivalent homogenous cloud"

- Page 6, line 27: "simulated 3D reflectances are lower than the 1D ones, the retrieved optical thickness is an effective optical thickness, lower than the averaged one (Figure 2)" -> "reflectances off of a heterogenous cloud are lower than the reflectances off of an equivalent homogenous cloud with the same (mean) COT, leading to an effective optical thickness, which is lower than the mean optical thickness."

=============== Page 7 (contd):

16: "consequences on" -> "consequences for"

19: "As previously" -> "As explained before"

21: "Figure 3" -> "Figure 2"

22: "saturate" -> "level off"

24: "comparing to" -> "compared to"

24: "reflectances fiels" -> "reflectance fields"

26: "facing to the sun" -> "facing the Sun"

27-29: If 41%, 52%, and 38% of the pixels are considered "a large part", aren't the remaining 59%, 48%, 62% even larger parts? I don't understand.

31: "shadows area" -> "shadow areas"

32: "reflectances pictures" -> "reflectance images"

============== Page 8:

3: "we plotted" -> "we plot"

3: "as function of" -> "as a function of"

5: "cloud appear" -> "clouds appear"

6: "cloudbow directions" -> "cloudbow direction"

7: "As for total reflectances" -> "Similar to the total reflectances"

8: "cloud respectively" -> "cloud, respectively"

15: "are consequently" -> "is consequently"

18: "become thus" -> "thus become"

20-21: "thus quite important" -> "thus important"

22: "no present" -> "not present"

26: "surnumerary" -> "supernumerary" Fix this throughout paper.

32: "7.5% error" and "5% error" Errors in what?

============== Page 9:

1-2: "As previously, we use again" -> "We use again"

3: "differences on" -> "differences in"

3: "quite large" -> "large"

5: "table" -> "Table"

6: "surnumerary" -> "supernumerary"

8: "wavelength are" -> "wavelengths are"

9: "that, the" -> "that the"

10: "more important" What does it mean?

11: "does even not" -> "does not even"

19: "in forward" -> "in the forward"

20: "that cloud" -> "that the cloud"

26: "Figure 6" -> "Figure 5"

28: "impacted but the" -> "impacted by the"

30: "3D clouds effects" -> "3D cloud effects"

33: "Figure 5" -> "Figure 6" I propose to swap Figures 5 and 6, because currently Fig. 6 is mentioned before Fig. 5. Also swap all the references to Figs. 5 and 6.

============== Page 10:

3: "vary with" -> "varies with"

5: "Figure 5" -> "Figure 6"

7: "biomass-burning layer" -> "biomass-burning aerosol layer"

10: "in function" -> "as a function"

11: "Figure 6" -> "Figure 5"

12: "heterogeneities effects" -> "heterogeneity effects"

13: "section" -> "Section"

17: "plane parallel" -> "plane-parallel"

25: "between 60" -> "between scattering angles of 60"

26: "which is comparable to" -> "which resembles"

28: "ones at 60 deg" -> "ones at 60 deg (see Fig. 2)"

28-29: "3D effects ... because of the plane-parallel bias" So far 3D effects was used to refer to brightening / shadowing and not for plane-parallel bias. I would not use 3D effects in this sentence, because this is really about plane-parallel bias.

29: "corresponds to" -> "resembles"

31: "at the two" -> "at two"

31-32: "corresponds in the retrieval to the best-selected model" What does this mean? In the next 2 sentences it looks like the Angstrom exponent is directly calculated instead of taken from a best-selected model.

32: "close for" -> "similar for"

33: "Angstrom" Add dots on "o". Fix this throughout paper.

============== Page 11:

1: "consists in" -> "consist in"

2: "angstrom" -> "Angstrom" incl. dots

5: "is 7 so above" -> "is 7, which is above the homogeneity limit for POLDER."

3-5: This belongs in Section 2 to put the synthetic clouds into perspective. Apparently, the fractional cloud with stdev(COT)=7 exceeds POLDER's homogeneity limit of 5. The

fractional cloud also gives the worst results compared to the flat and bumpy clouds. I think it would be good if the choice for stdev(COT)=7 would be justified in Section 2, and if at least rough numbers are given for how the results compare to a similar fractional cloud with stdev(COT)=5.

8: "retrieve values" -> "retrieved values"

8: "of optical" -> "of cloud optical"

9: "effective radius" -> "droplet effective radius"

10: "POLDER radiometer" -> "POLDER radiometer and polarimeter"

18: "yields to" -> "yields"

20: "to the shadowing" -> "to shadowing"

20: "maximal ? bias" ?!?

25: "lead to" -> "leads to"

26: "reduced the" -> "reduce the"

27: "are not too much affected" -> "ard hardly affected"

28: "surnumerary" -> "supernumerary"

32: "retrieve non-negligible" -> "retrieves non-negligible"

32: "value when" -> "values when"

32: "only a limited range of scattering angles is available" -> "only larger scattering angles are available" Important difference!

============== Page 12:

3: remove "" around "shadowing effects"

4: "and since the information of the" -> "and the"

10: "also obviously" -> "also"

14: "platform than the" -> "platform as the"

============== Page 17:

40 deg is missing in top row

============== Page 18:

1: "Fraction-al" -> "Fractional"

CTOP: Max(z) for the cloud top height of the 3D clouds does not seem like a representative value to me. See Fig. 1: realistic values are closer to 1.2 (I can only see y=3.5 but the realistic value should be computed from all y). Same for min(x) for bottom. Better values should be used, or at least the retrieval results should be compared to more than just max(z).

Caption: "Effective radius" -> "Cloud droplet effective radius"

============== Page 19:

"Angstrom" add dots

============== Page 21:

"Figure 2" -> "Figure 3"

6: "(1D) COT" -> "(1D) COT."

Plot symbols all look alike. (Symbol missing in "1D sza20".)

Hard to distinguish different shades of blue

============== Page 22:

"Figure 3" -> "Figure 2"

Minus sign in titles (e.g. Rtot - Cloudbow) is confusing.

5: "the forward scattering" -> "the cloudbow scattering"

I don't understand the exact meaning of the yellow color. And what does the black/grey mean?

============== Page 23:

Hard to distinguish two shades of blue.

4: "Polarized reflectances" -> "Polarized reflectance"

4: delete "(SZA=60 deg)

4: "in function of the scattering" -> "as a function of scattering"

6: "solar incidence" -> "solar zenith angle"

============== Page 24:

"Figure 5" -> "Figure 6"

============== Page 25:

"Figure 6" -> "Figure 5"

"reflectances in function" -> "reflectance as a function"

"solar incidence" -> "solar zenith angle"

Hard to distinguish shades of blue and red.

---

## Referee Comment (RC2) · Anonymous Referee #2 · 31 Jan 2018

The paper investigates errors due to cloud heterogeneity on operational retrieval algorithms developed for the POLDER radiometer. The methodology is appropriate: In a first step, realistic artificial cloud fields are generated using the model 3DCLOUD. Three cloud types with the same mean optical thickness are generated: a flat cloud, a bumpy cloud and one with fractional cloud cover. In a second step a 3D Monte Carlo radiative transfer model is applied to generate synthetic POLDER observations (simulations on fine spatial resolution are averaged over 7km x 7km to mimic POLDER pixels). The third step is to apply the operational POLDER algorithms on the synthetic data and compare the results with the known truth (artificial cloud fields). The following parameters are investigated: cloud optical thickness, cloud albedo, cloud top pressure,

aerosol optical thickness above cloud, and cloud size distribution parameters (effective radius and effective variance). The results show that all parameters except the size distribution parameters are highly biased compared to the truth.

The study is a very important validation of the POLDER algorithms, since it provides error estimates of the retrieved parameters due to cloud heterogeneity. The results show, that cloud heterogeneity can not be neglected and it should be taken into account in future retrieval algorithms.

The paper is well structured and quite well written. I recommend to publish the paper after revision (see comments and technical corrections).

Comments:

===========

- There are other studies related to this one by A. Stap et al. They have investigated the errors due to cloud heterogeneity on aerosol retrieval algorithms for partially cloudy scenes, also developed for the POLDER radiometer. These could be mentioned in the introduction.

F. A. Stap, O. P. Hasekamp, C. Emde, and T. Röckmann. Multiangle photopolarimetric aerosol retrievals in the vicinity of clouds: Synthetic study based on a large eddy simulation. Journal of Geophysical Research: Atmospheres, 121(21):12914-12935, 2016. 2016JD024787.

F.A. Stap, O.P. Hasekamp, C. Emde, and T. Röckmann. Influence of 3D effects on 1D aerosol retrievals in synthetic, partially clouded scenes. J. Quant. Spectrosc. Radiat. Transfer, 170:54 - 68, 2016.

- Since the retrieval errors due to cloud heterogeneity are large, the conclusion of the study should be that one should develop new retrieval algorithms, which somehow consider cloud heterogeneity. I miss this conclusion in the introduction and/or conclusions section.

[Figure]

Steps in this directions are presented in the following papers:

W. Martin, B. Cairns, G. Bal, Adjoint methods for adjust- ing three-dimensional atmosphere and surface properties to fit multi-angle/multi-pixel polarimetric mea- surements, J. Quant. Spectrosc. Radiat. Transfer 144 (2014) 68–85 doi:10.1016/j.jqsrt.2014.03.030

W. G. Martin, O. P. Hasekamp, A demonstration of ad- joint methods for multi- dimensional remote sensing of the atmosphere and surface, J. Quant. Spectrosc. Ra- diat. Trans- fer 204 (Supplement C) (2018) 215 – 231 doi:10.1016/j.jqsrt.2017.09.031

A. Levis, A. Aides, Y. Y. Schechner, and A. B. Davis, Airborne Three-Dimensional Cloud Tomography. In Proceedings of the IEEE International Conference on Computer Vision 2015 (ICCV15), pp. 3379-3387 (201 5). Available online at: http://www.cv-foundation.org/openaccess/content_iccv_2015/html/Levis_Airborne_Three-Dimensional_Cloud_ICCV_2015_paper.html

A. Levis, Y. Y. Schechner, and A. B. Davis, Multiple-Scattering Micro-physics Tomography. In Proceedings of the 30th IEEE/CVF Conference on Computer Vision and Pattern Recognition (CVPR17). Available onli ne at: http://openaccess.thecvf.com/content_cvpr_2017/papers/Levis_Multiple-Scattering_Microphysics_Tomography_CVPR_2017_paper.pdf

- The core of the study, the 3D radiative transfer (RT) model 3DMCPOL, is not de-scribed (only reference Cornel et al. 2010 is given). There should be a brief description on which methodology is used to solve the vector radiative transfer equation and also on the accuracy. Also later, in the results section, it is not mentioned, how accurate the radiative transfer simulations are. Can we trust the RT results, has the model been validated? The first paragraph in section 2 provides a short description of the cloud model 3DCLOUD; I would expect a similar description for 3DMCPOL.

- p5, l4: "To remain consistent with assumptions made within POLDER operational

algorithm, an oceanic surface with a wind speed of 7 m.s-1 is included for total re-flectances while a black surface is included for polarized reflectances." -> This is an odd assumption. I think that this could introduce large errors, because the sun-glint is highly polarized. Why is the surface inconsistently included in the POLDER oper-ational algorithm? Is there any document where this assumption is justified. Please explain/discuss this issue.

- p5, l17: "Note that in the three cases, the operational algorithm retrieves a cloud cover equal to one." -> can the operational algorithm retrieve cloud cover different from one? If yes, why does it not work for the fractional cloud?

- p.6, l1: "That confirms that heterogeneity parameters can be at first order used to characterize plan-parallel bias" -> could the heterogeneity parameter be derived from observations?

- p.6, l31: "Contrarily, using 1D cloud radiative model in the inversion and in the direct computation as it is done in the operational algorithm, is coherent and leads to a sound cloud albedo. The plane-parallel bias is indeed almost canceled."

This sounds as if the operational algorithm would retrieve a good cloud albedo, but it does of course not. The reality always "uses" a 3D radiative transfer model, so retrieval algorithms based on 1D RT models are always inconsistent and yield wrong results.

- p7, l1: "Albedos are simulated simply by summing the proportion of the Monte-Carlo photons going up at the top of atmosphere." -> This is then not the cloud albedo but the total albedo, since it includes also contributions from molecular scattering and surface reflection, right?

- p8, Sec4.2: The effective variance retrieval uses the amplitude of the surnumerary bows. The aerosol above cloud retrieval (Sec 4.3) obtains information about AOT from the attenuation of the cloud bow. If effective variance and AOT above cloud both influ-ence the amplitude of the cloudbow region, how does the retrieval distinguish between

higher AOT and narrower size distribution? Does the amplitude also depend on cloud optical thickness?

- Sec. 4.2: Is the optimal estimation method a good approach for Reff/Veff retrieval based on the polarization of the cloudbow region? You write that the radiance does not fit very well, so that the retrieval does not converge, although the retrieval of the size distribution parameters is very accurate. I would think that the retrieval should not minimize the fit to radiances but it should only fit the position of the cloudbow and its amplitude. This could be realized using an optimal estimation approach but may be a simple lookup-table method would also work well. Somehow the retrieval should provide a criterion, whether it provides good results or not, here the cost function is not a good number for the quality of the retrieval.

- Sec. 4.2: "For the misrepresentation of 3D effects, we add 7.5% error in the cloudbow direction and 5% elsewhere." -> how are these errors estimated? Please justify.

- Table 3: I can not believe that for SZA=40° the difference between true and retrieved AOT and Angstroem coefficient (here also SZA=20°) is exactly 0.0 (with 3 digits accuracy). Please explain why it is exactly the same.

Technical corrections:

===================

p2, l27 ff.: "Indeed, for oblique ..." -> the sentence is too long and confusing. It is not clear whether higher optical thickness is retrieved when the cloud is homogeneous or inhomogeneous. Or when the retrieval assumes a homogeneous or inhomogeneous cloud. Please rewrite.

p2, l30 ff.: "3D effects, which depends " -> which depend

p4, l17: "Wavelet"-> wavelet

p4, l23ff: "... fractional cloud has a fractional cloud cover fixed to 0.76 and a heterogeneity parameter equal to 1.12 if the heterogeneity parameter (optical standard deviation over the mean optical thickness) is computing including the zeros or 0.95 if it is computing only with the cloudy pixels." -> Sentence is not clear after "1.12", please rewrite.

p5, l2: "where I are" -> is

p5, l2: I think that the dots in the unit W.m-2.sr-1 are not AMT standard.

p6, l4: "differences are important according to the viewing direction." -> "differences highly depend on the viewing direction."

p6, l23: "from every viewing angles"-> "from all viewing angles"

p6, l30: "lower than the ones ..." -> "lower than the one ..." (cloud albedo)

p6, l32: "coherent"-> "consistent"

p7, l32: "shadows area" -> "shadow area"

p8, l3: "7x7km" -> "7kmx7km resolution"

p8, l4ff: text refers to Figure 4a and Figure 4b, but (a) and (b) is not included in Figure 4.

p8, l11: "appear differently according to ... " -> "appear differently depending on"

p8, l17: "At 670 nm, the polarized reflectance in the shadow part is only slightly enhanced by the molecular scattering but sufficiently compared to 865 nm." -> I do not understand what is meant by "sufficiently" in this sentence.

p8, l22: "no present" -> "not present"

p9, l3: "bias" -> "biases"

p9, l10: "much more important" -> "much larger"

p9, l32: Please define "Rayleigh pressure"

p10, l10: "polarized reflectances in function of the scattering angles ... " -> "polarized reflectances as function of the scattering angles ... "

p11, l8: "retrieve values" -> "retrieved values"

p11, l15: "In the case of the optical thickness and high solar illumination (SZA=20°), we obtain ..." -> "For high solar elevation (SZA=20°) the optical thickness retrieval yields ..."

p11, l17: "For POLDER, it leads to retrieve optical thicknesses underestimated by 10 or 35%" -> "For POLDER, the retrieved optical thicknesses are underestimated by 10 or 35% ..."

p11, l22: "cloud heterogeneities effects " -> "cloud heterogeneity effects"

p11, l26: What is meant with "negative polarization"?

---

## Referee Comment (RC3) · Z. Zhang (Referee) · 4 Feb 2018

Review on "Cloud heterogeneity effects on cloud and aerosol above cloud properties retrieved from simulated total and polarized reflectances" by Céline Cornet et al.

Summary: This paper presents a hypothetical study of how 3D radiative transfer effects influence the retrieval of cloud properties and above-cloud aerosol properties based on the multi-angular polarimetric observations. The study starts with forward radiative transfer simulations of POLDER observations for synthetic scenes with absorbing aerosols overlying fractal clouds. Then the simulated radiance and polarization observations are given to a retrieval-simulator to retrieve cloud properties and above-cloud

aerosol properties. Note that in the forward radiative transfer simulations are based on 3-D radiative transfer model, while the retrieval processes are based on the 1-D radiative transfer theory. Therefore, the differences between the retrieval results and the original clouds/aerosols properties reflect the influences of 3-D radiative effects on the retrievals. The results indicate that the microphysics of clouds are less sensitive to the 3-D effects, while other clouds and aerosol properties are more susceptible to 3-D effects.

The topic of this paper is suitable for AMT. The paper is the first one, as far as a I know, to discuss the impacts of 3-D effect on above-cloud aerosol property retrievals. In this regard, it is very important. However, I feel that the current version needs significant improvements before it can be accepted for publication. My major comments and suggestions are listed below.

Major comments/suggestions: " From my perspective, the largest contribution of this paper is that it advances our understanding of how 3D effects influence the retrieval of polarimetric based remote sensing of above-cloud aerosols. However, there is almost no mention about above-cloud aerosols, e.g., their occurrence frequency, global distribution, climate importance, remote sensing methods to retrieve their properties. The background information on above-cloud aerosols is important for the readers to appreciate the importance of this paper. By now, there is a significant volume of literature on this topic, for example, Chand et al. (2009); Zhang et al. (2016);

" In this study, the radiative transfer simulations are done at very high spatial resolution, 50 m. although the results are averaged to 7km to "mimic the radiometer measurements and applied the POLDER operational algorithm". Only retrievals at the 7km are presented and analyzed. The reason for the spatial average understandable. But the high-resolution radiative transfer and retrieval results (if any) should also be presented and analyzed for a couple of important reasons. First of all, the 3-D effects are highly dependent on the spatial scale. At small scale (e.g., 50m) the violation of independent pixel approximation (i.e., smoothing, illuminating and shadowing effects) is more

important, while at coarser resolution (e.g., 7km) the plane-parallel bias is more important, as pointed out in many previous studies including Zhang et al. (2012). Therefore, the high-resolution results, in combination with the low-resolution results, are very important for us to gain a comprehensive understanding of the problem. Second, the high-resolution results are very relevant to air-borne instruments, such as RSP and HARP. These instruments have been employed in the recent ORACLES field campaign. These air-borne instruments have spatial resolution on the order of 100m. So the results in this paper are highly relevant. Therefore, I strongly suggest the authors add some results and discussion on the high-resolution radiative transfer and retrieval results.

" This paper focuses on the polarimetric remote sensing technique. But it is somewhat disappointing that there is no discussion on the spectral methods for above-cloud aerosol retrievals (e.g., Jethva et al. 2013 and Meyer et al. 2015). As far as I understand, the radiative transfer and retrieval framework used in this study can be easily extended to the spectral method. I'd encourage the authors to take this opportunity to look into the 3-D effects on spectral based above-cloud aerosol retrievals. But I will leave this to the authors to decide whether they will do this in this study or future work.

" What is not clear from the current paper is how much the retrieval error is due to the 3D effects and how much is due to retrieval algorithm uncertainty. For example, POLDER has a coarse angular resolution and it seems to me this is partly the reason why the above cloud AOD retrieval error is large in Table 3. Also, in the retrieval process based on the Waquet et al. (2013), how much a priori information is given to the retrieval algorithm? Does the retrieval algorithm know, for example, the single scattering albedo of the above-cloud aerosol at each wavelength? In reality, the algorithm certainly does NOT know the aerosol properties. Some discussions are needed to clarify how aerosols are treated in the Waquet et al. (2013) retrieval algorithm and justify the treatment.

" Related to the last point, the AOD retrieval error could be put into a more meaningful

context. For example, what is the relative error in AOD retrieval if the assumption of single-scattering albedo of aerosols is wrong in the retrieval algorithm? How is this error compared with the 3-D effects? Such comparison will help us understand the relative importance of 3-D effects in comparison with some other error sources in the retrieval.   Chand, D., R. Wood, T. L. Anderson, S. K. Satheesh, and R. J. Charlson (2009), Satellite-derived direct radiative effect of aerosols dependent on cloud cover, Nature Geoscience, 2(3), 181-184, doi:10.1038/ngeo437.

Jethva, H., O. Torres, L. A. Remer, and P. K. Bhartia (2013), A color ratio method for simultaneous retrieval of aerosol and cloud optical thickness of above-cloud absorbing aerosols from passive sensors: Application to MODIS measurements, IEEE TRANS-ACTIONS ON GEOSCIENCE AND REMOTE SENSING, 51(7), 3862-3870.

Meyer, K., S. Platnick, and Z. Zhang (2015), Simultaneously inferring above‐cloud absorbing aerosol optical thickness and underlying liquid phase cloud optical and microphysical properties using MODIS, Journal of Geophysical Research-Atmospheres, 120(11), 5524-5547, doi:10.1002/2015JD023128.

Zhang, Z., K. Meyer, H. Yu, S. Platnick, P. Colarco, Z. Liu, and L. Oreopoulos (2016), Shortwave direct radiative effects of above-cloud aerosols over global oceans derived from 8 years of CALIOP and MODIS observations, ACP, 16(5), 2877-2900, doi:10.5194/acpd-15-26357-2015.

---

## Referee Comment (RC4) · F. Xu (Referee) · 11 Feb 2018

Spatial distribution of aerosol and cloud microphysical properties and aerosol/cloud interaction are highly concerned by our community. In this context, the theoretical study performed by Cornet et al. on cloud heterogeneity effects on cloud and aerosol above cloud remote sensing is important and fits very well the direction of remote sensing algorithm development.

I have no questions about the tools (including the Monte Carlo polarized radiative transfer model and POLDER cloud and above-cloud aerosol retrieval algorithm employed in this study) as they have been well developed and validated in LOA over the last two

decades. Beyond the opinions of the other three reviewers, I have a few comments on the technical aspects:

1) The authors may double check Eq.(1) as it is more like a definition for bidirectional reflectance factors (BRF) instead of for "total reflectance"? In addition, to define polarized reflectance, it is better to use sqrt(Qˆ2+Uˆ2+Vˆ2) instead of "I" in Eq(1) for clarity. 2) Does the AOT retrieval closure test use the simulated signals from the whole scattering angular range from 60 to 180 degree ? It can be observed from Figs. 4 and 6 that the 3D impact is more remarkable in the scattering angular ranges from 60 to 80 degrees and from 160 to 180 degrees. What if the authors try doing the aerosol optical thickness (AOT) retrieval using the signals from 80-160 degrees range only (where 1D RT apparently has less plane-parallel bias) and re-evaluating the 3D impact on AOT retrieval ? I assume the aerosol information residing in this reduced angular range may be good enough for AOT retrieval (and may result in reduced error). 3) For solar incidence angles 20 and 40 degrees, the cloudbow signals (e.g. in the principal plane) should appear in two sides around incidence ray. And their magnitudes should be somehow different. But such a difference is not observed in Figs. 4-6. Is this due to the signals at the same scattering angles are just averaged regardless of the difference in viewing angles ? It may be more clear if the authors plot both of them in those figures. 4) It may be necessary to describe a little more on the criterion for setting 50 m as the small scale (pixel scale). Is this set up due to the sufficiency in ensuring a) representativeness of cloud microphysical property variation and/or b) accuracy of cloud signals in a certain scale ?

———————————————

---

## Author Comment (AC1) · 2 Jun 2018

**Reply to G. Van Harten**

The authors would like to thank G. Van Harten for its valuable comments and suggestions that improve greatly the paper. We also acknowledge him deeply for his carefully reading and time spent to list all the point concerning grammatical and vocabulary errors. That increases a lot the readability of the paper.

Please, find below the point-by-point answers to the specific comments.

**Specific comments**

**To draw the right conclusions on the effects of heterogeneity, it is important that the clouds are as similar as possible, except for their heterogeneity. Listed below are comments related to the choice of simulation parameters:**

**Page 4, line 23-24: Why are the rho's for the fractional cloud not closer to 0.6 for better comparison to the flat and bumpy cloud?**

The aim of the paper is to study cloud heterogeneity effects for typical clouds. We thus generated clouds according to typical values of heterogeneity parameters. However, the choice of these values is not easy as the estimation of the heterogeneity parameter is not straightforward and depends on different parameters such as the type of measurement (radiometric data, radar/lidar data, airborne in-situ data), on the measured quantity (optical depth, liquid water content) and on the spatial resolution and scale used to compute it. Shonk et al.(2010) made a review of the different definitions and values that can be found in the scientific literature. We chose to follow the values obtained by Barker et al.(1996) from Landsat as the spatial resolution of the instrument (50m) is close to the spatial resolution of our simulations. Barker et al..(1996) found values between 0.2 and 0.8 for overcast stratocumulus clouds and 0.6 to 2.3 for small cumulus or broken clouds.

To explain our choice, we modified the paragraph page 7 as:

We created two stratocumulus clouds and one cumulus cloud. The latter is the result of instabilities of the boundary layer and lead to fractional cloud cover and larger heterogeneity parameter (Kawai and Teixeira, 2011). The flat and bumpy clouds representing overcast stratocumulus clouds have the same heterogeneity parameter across the 140x140 pixels with $\rho = 0.6$. The cumulus cloud has a fractional cloud cover equal to 0.76 and a heterogeneity parameter equal to 1.12 setting clear sky pixels to null values (0.95 if computed only with the cloudy pixels). These values are typical values obtained from Landsat data (Barker et al., 1996) for stratocumulus and cumulus clouds.

**Page 4, line 28: How are the flat and bumpy clouds parameterized? What are the settings for cloud top height, etc?**

The users do not prescribe explicitly the cloud top height nor the bumps structures. Indeed, they are the result of the numerical simulation using basic cloudy atmospheric numerical equations (first step of the 3DCLOUD algorithm). This numerical simulation is driven by the assimilation of the meteorological vertical profiles prescribed by the user.

To explain better how the clouds are generated, we add Figure 1 in the paper and this paragraph:

Figure 1 shows the vertical profiles of potential temperature and of vapor mixing ratio prescribed in this study to generate the three cloud fields. Globally, the vertical profiles of potential temperature and vapor mixing ratio give the cloud position. The mean cloud top height is mainly determined by the height where the potential temperature increases and the vapor mixing ratio decreases. Cloud top height fluctuations (shapes of top bumps) are mainly the result of the intensity of the vertical gradient of the potential temperature and vapor mixing ratio.

[Figure]

*Figure 1:Vertical profiles of potential temperature and of vapor mixing ratio prescribed in this study to generate the flat stratocumulus (circle), the bumpy stratocumulus(point) and the cumulus (star) cloud fields.*

**Page 5, line 6: Why a black surface for polarized reflectances? The surface seems important in particular for the fractional cloud. At certain angles it can be very bright in polarization (sun glint).**

The operational algorithm using polarized reflectances assumes a black surface because the multi-angularity of POLDER allow to not use the directions close to the sun glint where polarized reflectances can be high. In the other directions the polarized ocean surface reflection is almost null (black).

We add page 7 :
Indeed, for retrieval using polarized reflectances, the multi-angular ability of POLDER provides the advantage of not using the directions close to the sun-glint where polarized reflectances can be high.

**Page 5, line 15; Page 18, CTOP: Max(z) / min(z) for cloud top height / bottom of the 3D clouds does not seem like a representative value to me. See Fig. 1: realistic values are closer to 1.2 (Fig. 1 only shows y=3.5 whereas the realistic value should be computed from all y). Better values should be used, or at least the retrieval results should be compared to more than just max(z).**

You're right that max(z) was not a representative value for cloud top height. Following your comment, we computed the mean cloud top height. The table 2 was changed accordingly as well as comments page 9 :
In table 2, we report the mean cloud top height for each heterogeneous cloud and the retrieved value. The 1D homogeneous values used for control was set the intermediate mean cloud top altitude. We note slight difference about -4 hPa (+ 37m) between input and 1D retrieval, which reveals small differences between the radiative transfer codes used for the simulation and for the retrieval. Differences between

3D and 1D are however much larger, especially for the bumpy and fractional cloud with values of +62hPa (-550m) and +45hPa (-390m).

**Page 11, line 3-5: This belongs in Section 2 to put the synthetic clouds into perspective. Apparently, the fractional cloud with stdev(COT)=7 exceeds POLDER's homogeneity limit of 5. The fractional cloud also gives the worst results compared to the flat and bumpy clouds. I think it would be good if the choice for stdev(COT)=7 would be justified in Section 2, and if at least rough numbers are given for how the results compare to a similar fractional cloud with stdev(COT)=5.**

As explained before, we chose typical value of heterogeneity parameter corresponding to the cumulus cloud. We apply the aerosol above cloud algorithm to the worst case, which can be seen as the upper limit of the possible error on retrieved AOT. Afterwards, when we computed the stdev(COT) from the 1km pixels real COT (different to the retrieved one), we found a value slightly above the limit fixed arbitrary in the algorithm. For computational time reason, that is not possible to modify the cloud case in order to have stddev(COT)=5 and we do not think that the results would be a lot different. We modified the paragraph page 15-16.

For the fractional cloud of this study, we checked the standard deviation value computed from the input cloud optical thickness (different from the retrieved one) and found 7. It is slightly above the homogeneity limit fixed in the aerosol above cloud algorithm developed for POLDER (Waquet et al., 2013). The results presented here for aerosol above cloud retrieval can thus be seen as an upper limit for the operational algorithm.

**Detailed comments:**

Text has been modified according to all the detailed comments addressed in the review. See the joined file with tracking changes. Note that some corrections were already made in the preview phase.
The authors acknowledge deeply G. Van Harten for his careful attention and the time spent to do it.

Please find below answers to comments, which require more precise answers and were not reported in the specific comment section.

**26: "which is supposed, in real cloud, to be more important than" Reference? How much more important? Which retrieval parameters are affected?**

We add as the reference (Magaritz-Ronen L. et al., 2016) which explores the mechanisms leading to low horizontal variability of effective radius at the top of the cloud and gives many others references in the introduction. The maximum variability is estimated to be of order of 10%.

**18: "algorithm retrieves" Does it really retrieve cloud cover, or should it say "algorithm assumes"?**

Right, the cloud cover is an output of the algorithm for the super pixel POLDER but here the term "retrieve" is not adequate. We removed the sentence

**P7 :**
**I found the paragraph about albedo hard to read. If I understand correctly, the train of reasoning is: - In order to retrieve cloud albedo, the POLDER retrieval algorithm first retrieves COT from multi-angle reflectances, then does a forward computation from COT to albedo. - A heterogeneous cloud has lower reflectance than a homogenous cloud with the same (mean) cloud optical depth, due to plane-parallel bias. - The POLDER algorithm will thus retrieve a COT that is too low. - From that too-low COT, the POLDER forward computation will produce an albedo that is also too low if the cloud were really a homogeneous cloud, but since it is really a heterogeneous cloud with a**

**lower albedo due to plane-parallel bias, the POLDER-retrieved albedo is actually very close.**

You understand well.

**- To compare this retrieval result to the "truth", the actual albedos are directly calculated from the 3DMCPOL radiances. It would be helpful if this could be explained in a more direct way, e.g. by preparing the reader by summarizing this at the beginning of the paragraph, before going into the details.**

**Other suggestions for textual changes:**
**- Page 6, line 26: "3D reflectances and from 1D reflectances are not comparable" -> "a heterogeneous cloud are not the same as the ones retrieved from an equivalent homogenous cloud"**
**- Page 6, line 27: "simulated 3D reflectances are lower than the 1D ones, the retrieved optical thickness is an effective optical thickness, lower than the averaged one (Figure 2)" -> "reflectances off of a heterogenous cloud are lower than the reflectances off of an equivalent homogenous cloud with the same (mean) COT, leading to an effective optical thickness, which is lower than the mean optical thickness."**

Following the above advices of the reviewer and hoping to be clearer, we rephrase the paragraph about albedo as:
The assessment of cloud heterogeneity effects on cloud albedo is realized by comparing the retrieved POLDER algorithm albedos with the ones directly computed with the 3DMCPOL radiative transfer model identified as the true one. Direct comparison of retrieved albedos values from homogeneous or from the heterogenous clouds as done for other parameters are not suitable for cloud albedo. Indeed, the plane-parallel bias leads to reflectances off of a heterogenous cloud lower than the reflectances off of an equivalent homogenous cloud with the same (mean) COT. The retrieved optical thickness is lower than the mean optical thickness of 10 (Figure 4). Using it to recompute the albedo in the POLDER algorithm leads to a too low value comparing to the albedo of the equivalent homogeneous cloud. Contrarily, using 1D cloud radiative model in the inversion and in the direct computation as it is done in the operational algorithm, is consistent and leads to a sound cloud albedo. The plane-parallel bias is indeed almost cancelled.

**P7 : 27-29: If 41%, 52%, and 38% of the pixels are considered "a large part", aren't the remaining 59%, 48%, 62% even larger parts? I don't understand.**

We wanted to highlight that 41%, 52% and 38% of reflectances pixels cannot be reproduced and explained with 1D radiative transfer but only with 3D radiative transfer. We add "and thus cannot be obtained with 1D radiative transfer simulation"

Barker, H. W., Wiellicki, B. A. and Parker, L.: A Parameterization for Computing Grid-Averaged Solar Fluxes for Inhomogeneous Marine Boundary Layer Clouds. Part II: Validation Using Satellite Data, J. Atmospheric Sci., 53(16), 2304–2316, doi:10.1175/1520-0469(1996)053<2304:APFCGA>2.0.CO;2, 1996.

Kawai, H. and Teixeira, J.: Probability Density Functions of Liquid Water Path and Total Water Content of Marine Boundary Layer Clouds: Implications for Cloud Parameterization, J. Clim., 25(6), 2162–2177, doi:10.1175/JCLI-D-11-00117.1, 2011.

Magaritz-Ronen L., Khain A. and Pinsky M.: About the horizontal variability of effective radius in stratocumulus clouds, J. Geophys. Res. Atmospheres, 121(16), 9640–9660,

doi:10.1002/2016JD024977, 2016.

Shonk Jonathan K. P., Hogan Robin J., Edwards John M. and Mace Gerald G.: Effect of improving representation of horizontal and vertical cloud structure on the Earth's global radiation budget. Part I: Review and parametrization, Q. J. R. Meteorol. Soc., 136(650), 1191–1204, doi:10.1002/qj.647, 2010.

Waquet, F., Cornet, C., Deuzé, J.-L., Dubovik, O., Ducos, F., Goloub, P., Herman, M., Lapyonok, T., Labonnote, L. C., Riedi, J., Tanré, D., Thieuleux, F. and Vanbauce, C.: Retrieval of aerosol microphysical and optical properties above liquid clouds from POLDER/PARASOL polarization measurements, Atmos Meas Tech, 6(4), 991–1016, doi:10.5194/amt-6-991-2013, 2013.

---

## Author Comment (AC2) · 2 Jun 2018

**Reply to the anonymous referee 2**

The authors thank the anonymous reviewer for his comments and suggestions to improve the paper. Please find hereafter our point-by-point responses to comments and suggested corrections. We also acknowledge him deeply for his carefully reading and the time spent to list all the point concerning grammatical and vocabulary errors. It increases a lot the readability of the paper.

**- There are other studies related to this one by A. Stap et al. They have investigated the errors due to cloud heterogeneity on aerosol retrieval algorithms for partially cloudy scenes, also developed for the POLDER radiometer. These could be mentioned in the introduction. F. A. Stap, O. P. Hasekamp, C. Emde, and T. Röckmann. Multiangle photopolarimetric aerosol retrievals in the vicinity of clouds: Synthetic study based on a large eddy simulation. Journal of Geophysical Research: Atmospheres, 121(21):12914-12935, 2016. 2016JD024787.**
**F.A. Stap, O.P. Hasekamp, C. Emde, and T. Röckmann. Influence of 3D effects on 1D aerosol retrievals in synthetic, partially clouded scenes. J. Quant. Spectrosc. Radiat. Transfer, 170:54 - 68, 2016.**

Thank you suggesting these very interesting publications. We add them in the introduction:
In case of partial cloudy scenes, shadow, cloud enhancement of the clear areas by neighboring clouds can modify the retrieved aerosol properties. Errors on the retrieved aerosol properties are dependent of the cloud distribution, optical thickness and spatial resolution (Stap et al., 2016a; Stap et al., 2016b).

And in the conclusion section :
Further that assessments of cloud heterogeneity uncertainties, more complex methods should also be developed to retrieve aerosol and cloud properties accounting for the cloud heterogeneities. Several theoretical or case studies have already been conducted. Some tends to mitigate cloud contamination for aerosol property retrieval (Davis et al., 2013; Stap et al., 2016b).

**Since the retrieval errors due to cloud heterogeneity are large, the conclusion of the study should be that one should develop new retrieval algorithms, which somehow consider cloud heterogeneity. I miss this conclusion in the introduction and/or conclusion section.**
**Steps in this directions are presented in the following papers:**
**W. Martin, B. Cairns, G. Bal, Adjoint methods for adjusting three-dimensional atmosphere and surface properties to fit multi-angle/multi-pixel polarimetric measurements, J. Quant. Spectrosc. Radiat. Transfer 144 (2014) 68–85 doi:10.1016/j.jqsrt.2014.03.030**
**W. G. Martin, O. P. Hasekamp, A demonstration of adjoint methods for multidimensional remote sensing of the atmosphere and surface, J. Quant. Spectrosc. Radiat. Trans- fer 204 (Supplement C) (2018) 215 – 231 doi:10.1016/j.jqsrt.2017.09.031**
**A. Levis, A. Aides, Y. Y. Schechner, and A. B. Davis, Airborne Three-Dimensional Cloud Tomography. In Proceedings of the IEEE International Conference on Computer Vision 2015 (ICCV15), pp. 3379-3387 (201 5). Available online at: http://www.cvfoundation. org/openaccess/content_iccv_2015/html/Levis_Airborne_Three-Dimensional_Cloud_ICCV_2015_paper.html**
**Levis, Y. Y. Schechner, and A. B. Davis, Multiple-Scattering Microphysics Tomography. In Proceedings of the 30th IEEE/CVF Conference on Computer Vision and Pattern Recognition (CVPR17). Available online at:**
**http://openaccess.thecvf.com/content_cvpr_2017/papers/Levis_Multiple,**
**Scattering_Microphysics_Tomography_CVPR_2017_paper.pdf**

We agree with the comment and add a paragraph in the conclusion:

Further that assessments of cloud heterogeneity uncertainties, more complex methods should also be developed to retrieve aerosol and cloud properties accounting for the cloud heterogeneities. Several theoretical or case studies have already been conducted. Some tends to mitigate cloud contamination for aerosol property retrieval (Davis et al., 2013; Stap et al., 2016b). Others aim to use 3D radiative transfer model to retrieve 3D cloud properties and hence account for some cloud heterogeneity effects. It requires then more complex inversion methods. Feasibility studies has been conducted using neural network method (Cornet et al., 2004, 2005), 3D tomography with a surrogate function (Levis et al., 2015, Levis et al. 2017) or adjoint method (Martin et al., 2014; Martin and Hasekamp, 2018). The latter two methods are very promising but have been developed in the framework of high resolution measurements (ten to hundred meters) involving no or small plane-parallel bias. They are so not directly applicable to POLDER/PARASOL measurements.

.

**The core of the study, the 3D radiative transfer (RT) model 3DMCPOL, is not described (only reference Cornel et al. 2010 is given). There should be a brief description on which methodology is used to solve the vector radiative transfer equation and also on the accuracy. Also later, in the results section, it is not mentioned, how accurate the radiative transfer simulations are. Can we trust the RT results, has the model been validated? The first paragraph in section 2 provides a short description of the cloud model 3DCLOUD; I would expect a similar description for 3DMCPOL.**

For the description of the 3DMCOL model, we add in section 2:
It is a forward Monte-Carlo model able to compute radiative reflected or transmitted Stokes vector as well as upwelling and downwelling fluxes in three-dimensional atmospheres. Initially, developed for solar radiation (Cornet et al., 2010), it was next extended to thermal radiation (Fauchez et al., 2014). To save time and for an accurate computation of reflectances, the local estimate method (Marshak and Davis, 2005) is used. Periodical boundary conditions at the horizontal domain limits are used. For highly peaked phase function, the potter truncation is implemented. Molecular scattering is computed according to the pressure profile. A heterogeneous surface can also be specified with Lambertian reflection, ocean or snow bidirectional function. The model participated and was improved during the Intercomparison of Polarized Radiative Transfer model (IPRT) on homogeneous cloud cases (Emde et al., 2015) and on 3D cloud cases (Emde et al., 2018).

Concerning the accuracy of the computations used is the paper, we add:
Simulations are run with a total of $10^7$ photons and $10^9$ photons for the homogeneous and heterogeneous clouds respectively. The Monte-Carlo uncertainties are estimated with the computation of standard deviation with 10 and 50 independent realizations of $10^6$ and $20.10^6$ photons for the homogeneous and heterogeneous cloud respectively. For the homogeneous case, the relative standard deviation is below 0.12% for the total reflectances and below 1.2% for the polarized reflectances. For the heterogeneous clouds, at 50m resolution, the mean relative standard deviation is below 1.3% for the total reflectances. For polarized reflectances at 50m, the mean relative standard deviation varies according to the angular geometry and is between 2% and 107% for very small reflectance values with an mean value of 23%. At 7km, as the reflectances are averaged, relative standard deviation values are much lower below 0.01% and 0.8% for total and polarized reflectances respectively.

**p5, l4: "To remain consistent with assumptions made within POLDER operational algorithm, an oceanic surface with a wind speed of 7 m.s-1 is included for total reflectances while a black surface is included for polarized reflectances." -> This is an odd assumption. I think that this could introduce large errors, because the sun-glint is highly polarized. Why is the surface inconsistently included in the POLDER operational algorithm? Is there any document where this assumption is justified. Please explain/discuss this issue.**

As already discussed in the reply to reviewer 1, the operational algorithm using polarized reflectances assumes a black surface because the multi-angularity of POLDER allow to not use the directions close

to the sun glint where polarized reflectances can be high. In the other directions the polarized ocean surface reflection is almost null (black).

We add page 7 :
Indeed, for retrieval using polarized reflectances, the multi-angular ability of POLDER provides the advantage of not using the directions close to the sun-glint where polarized reflectances can be high.

**p5, l17: "Note that in the three cases, the operational algorithm retrieves a cloud cover equal to one." -> can the operational algorithm retrieve cloud cover different from one? If yes, why does it not work for the fractional cloud?**

The cloud cover is an output of the algorithm for the super pixel POLDER but you are right as the pixel level the value can only be zero or one. We removed the sentence

**- p.6, l1: "That confirms that heterogeneity parameters can be at first order used to characterize plan-parallel bias" -> could the heterogeneity parameter be derived from observations?**

The heterogeneity parameter cannot directly be obtained for one reflectance measurement but it may be estimated from higher spatial resolution measurements. This is the idea of the sentence wrote in the conclusion section:
"The Multi-viewing, Multi-Channel, Multi-Polarization Imaging mission (3MI) that will fly on METOP-A SG as part of EUMETSAT Polar System after 2021, will have a spatial resolution of 4 x 4 km. The plane-parallel bias is thus expected to be lower than for the POLDER instrument. In addition, as 3MI will be on the same platform as the Visible Infrared Imager (VII), a multispectral radiometer with a resolution of 500 m, the correction of the plane parallel biases may be possible while the multi-angular capability of 3MI would help to detect the illumination and shadowing effects."

**p.6, l31: "Contrarily, using 1D cloud radiative model in the inversion and in the direct computation as it is done in the operational algorithm, is coherent and leads to a sound cloud albedo. The plane-parallel bias is indeed almost canceled." This sounds as if the operational algorithm would retrieve a good cloud albedo, but it does of course not. The reality always "uses" a 3D radiative transfer model, so retrieval algorithms based on 1D RT models are always inconsistent and yield wrong results.**

Using an homogeneous cloud model for the cloud optical thickness retrieval from real or 3D reflectances and also for the computation of the cloud albedo almost cancel the plan-parallel bias effect. The residual error is due to the non-linearity degree of the reflectances/albedo as a function of the cloud optical thickness and to the 3D effects such as illumination, shadowing or even smoothing effects for high resolution. The reviewer 1 also found this paragraph unclear, we rephrased it hoping to be clearer:
The assessment of cloud heterogeneity effects on cloud albedo is realized by comparing the retrieved POLDER algorithm albedos with the ones directly computed with the 3DMCPOL radiative transfer model identified as the true one. Direct comparison of retrieved albedos values from homogeneous or from the heterogenous clouds as done for other parameters are not suitable for cloud albedo. Indeed, the plane-parallel bias leads to reflectances off of a heterogenous cloud lower than the reflectances off of an equivalent homogenous cloud with the same (mean) COT. The retrieved optical thickness is lower than the mean optical thickness of 10 (Figure 4). Using it to recompute the albedo in the POLDER algorithm leads to a too low value comparing to the albedo of the equivalent homogeneous cloud. Contrarily, using 1D cloud radiative model in the inversion and in the direct computation as it is done in the operational algorithm, is consistent and leads to a sound cloud albedo. The plane-parallel bias is indeed almost cancelled.

**p7, l1: "Albedos are simulated simply by summing the proportion of the Monte-Carlo photons going up at the top of atmosphere." -> This is then not the cloud albedo but the total albedo,**

**since it includes also contributions from molecular scattering and surface reflection, right?**

Good point, it is indeed a misnomer. The total albedo including molecular scattering, cloud scattering and surface reflection is indeed computed. It is done in the same way for the LUT used in the POLDER algorithm (Buriez et al., 2005). We replaced *cloud albedo* by the terms albedo of a cloudy scene or albedo only.

**p8, Sec4.2: The effective variance retrieval uses the amplitude of the surnumerary bows. The aerosol above cloud retrieval (Sec 4.3) obtains information about AOT from the attenuation of the cloud bow. If effective variance and AOT above cloud both influence the amplitude of the cloudbow region, how does the retrieval distinguish between higher AOT and narrower size distribution? Does the amplitude also depend on cloud optical thickness?**

The POLDER "operational algorithm" for aerosol above cloud retrieval uses a specific retrieval strategy. The cloud bow is indeed used for above cloud aerosol retrievals only in case of dust particles above clouds. The magnitude of the primary cloud bow primarily depends on the cloud droplet effective radius and this parameter must be also estimated. Collocated cloud properties from MODIS at high resolution (1 km × 1 km) are used to characterize and to select the cloudy scenes within a POLDER pixel (6 km × 6 km at nadir). The MODIS cloud products are notably used in the POLDER "operational algorithm" to estimate the droplets effective radius. The magnitude of the primary cloud bow is only weakly impacted by the choice of the droplets effective variance and this parameter is then fixed to 0.06 in the "operational algorithm".

We added this paragraph in the manuscript :
The magnitude of the primary cloud bow primarily depends on the cloud droplet effective radius and this parameter must be also estimated or included in the retrieval process. Collocated cloud properties from MODIS at high resolution (1 km × 1 km) are used to characterize and to select the cloudy scenes within a POLDER pixel (6 km × 6 km at nadir). The MODIS cloud products are notably used in the "operational algorithm" to estimate the droplets effective radius. The magnitude of the primary cloud bow is only weakly impacted by the choice of the droplets effective variance and this parameter is then fixed to 0.06 in the "operational algorithm".

and this information concerning the test realized :
Note, that for the synthetic retrievals discussed here below, we assumed that the operational algorithm knows the effective radius and effective variance of the cloud droplets.

For cloud optical thickness larger than 3, the amplitude of the cloud bow does not depend on the cloud optical thickness. We added this precision in the manuscript :
The retrievals are restricted to cloudy pixels associated with cloud optical thicknesses larger than 3.0, since the polarized radiance reflected by the cloud layer is then saturated and does not depend anymore on the cloud optical thickness.

**Sec. 4.2: Is the optimal estimation method a good approach for Reff/Veff retrieval based on the polarization of the cloudbow region? You write that the radiance does not fit very well, so that the retrieval does not converge, although the retrieval of the size distribution parameters is very accurate. I would think that the retrieval should not minimize the fit to radiances but it should only fit the position of the cloudbow and its amplitude. This could be realized using an optimal estimation approach but may be a simple lookup-table method would also work well. Somehow the retrieval should provide a criterion, whether it provides good results or not, here the cost function is not a good number for the quality of the retrieval.**

Beside the computation cost, the optimal estimation approach was chosen because of its flexibility. We want to keep this in order to have the freedom of adding new measurements or parameters in the state vector (like a second scattering layer above cloud for example).

We agree with the reviewer, that with a large sampling, a retrieval using only the position of the cloud bow and surnumerary bow would be much powerful than the absolute polarized radiance. However, one difficulty with POLDER/PARASOL measurements is that, because of the angular sampling, we never get the exact position of the maximum. A small error in the position of the maximum turns in a very large error in the effective radius. Therefore this « maximum position method » might give worst results than using the absolute polarized radiance.

The cost function is just an indicator of the goodness of the convergence within the errors provided by the measurements and forward model, and is also used as a criteria to stop the iteration process. Because the cost function is a sum of the square of standard normal variables, and because we have assumed that the conditional probability function of the measurements knowing the true state vector follows a normal distribution, the cost function follow a Chi-square law. We can therefore use this low together with a hypothesis testing to determine whether the weighed distance between the forward model and measurement is acceptable for a given confidence. This is just a statistical criteria which is working pretty well. A good indicator of the quality of the retrieval is always difficult to define, but the cost function at least can help when something went wrong in the retrieval, and especially when the forward model is not able, because it is too simple, to reproduce the measurements behavior (in the presence of highly heterogeneous cloud, or in the presence of an aerosol/cirrus layer above the liquid cloud).

We completed the sentence:
For all clouds, even if differences in polarized reflectances are large in amplitude, the retrieval algorithm still capture the general angular features of the three wavelengths, which results of small errors on the retrieved effective radius and effective variance.

And add concerning the cost function:
It means that the forward model (homogeneous model) used for the retrieval does not allow matching perfectly the heterogeneous cloud reflectances used as input.

**- Sec. 4.2: "For the misrepresentation of 3D effects, we add 7.5% error in the cloudbow direction and 5% elsewhere." -> how are these errors estimated? Please justify.**

These errors were estimated in previous work (Waquet et al., 2013) with the computation of 3-D and 1-D polarized radiances of a stratocumulus cloud close to the flat cloud presented here. Excepted for reflectances close to zero, relative errors were under 5-8%. We add the reference in the text as previous computations made in (Waquet et al., 2013)

**- Table 3: I can not believe that for SZA=40_ the difference between true and retrieved AOT and Angstroem coefficient (here also SZA=20_) is exactly 0.0 (with 3 digits accuracy).**
**Please explain why it is exactly the same.**

We checked the results and they are good. The rapid algorithm used for operational retrieval is based on precomputed tables. In the two cases, homogeneous and fractional cloud, the best model that minimized the cost function is the same so we obtain the same AOT. However, the cost function is more important for the heterogeneous cloud. We add the RMSE value between the input and the recalculated reflectances the table 3 and this sentence.
For SZA=40°, the best model that minimized the cost function is the same for the homogeneous and fractional cloud. Differences for the retrieved AOT are negligible, but we note that the RMSE between the input and recalculated reflectances is slightly larger for the fractional cloud than for the homogenous one.

**Technical corrections:**
Text has been modified according to the technical correction addressed by the reviewer that we would like to thank again. See the track changes file for the details..

---

## Author Comment (AC3) · 2 Jun 2018

**Reply to Z. Zhang,**

The authors would like to thank Z. Zhang for its valuable comments and suggestions that lead to improve the paper. The different relevant questions made by Z. Zhang and also by the others reviewers led to rewrite almost integrally the section 4.3 concerning the heterogeneity impacts of aerosol above cloud retrieval. We hope that it is now clearer.

Please, find below answers to the comments
**Major comments/suggestions: " From my perspective, the largest contribution of this paper is at it advances our understanding of how 3D effects influence the retrieval of polarimetric based remote sensing of above-cloud aerosols. However, there is almost no mention about above-cloud aerosols, e.g., their occurrence frequency, global distribution, climate importance, remote sensing methods to retrieve their properties. The background information on above-cloud aerosols is important for the readers to appreciate the importance of this paper. By now, there is a significant volume of literature on this topic, for example, Chand et al. (2009); Zhang et al. (2016);**

We agree that aerosol above cloud retrieval is of main importance and that represents a significant part of our paper. However, it is not the unique topic of the paper. It is more generally focused on cloud heterogeneity effects on POLDER measurements and parameters that can be retrieved from them. Off course, many previous papers have already studied the cloud heterogeneity effect on optical thickness. The main differences here is that we focus on the POLDER instrument algorithm, which has a lower resolution but takes advantages on its multi-angularity. More original is the study of the cloud heterogeneity effects on polarized reflectances and on the parameters that can be retrieved from it. Aerosol above cloud optical thickness is one of them as well as effective radius, effective variance and cloud top pressure.

But, we agree than the importance and improving our knowledge of aerosols in cloudy scene is not enough presented in the introduction. To improve it, we add some sentences and references including the one given by the reviewer.
In the introduction section we add several paragraphs :
In addition, absorbing aerosol above clouds can generate a positive direct radiative forcing (i.e. warming), that is currently not well quantified, and modify the properties of the below cloud layer (Chand et al., 2009, Wilcox, 2010 and Costantino et Bréon, 2013).

and :
Concerning aerosols, spaceborne active instruments, such as the lidar CALIOP are dedicated tools to detect multi-layer situations and to retrieve Aerosol Above Cloud (AAC) properties (Young and Vaughan, 2009, Hu et al., 2007, Chand et al., 2008) and were used for climate studies (Zhang et al., 2016). Passive measurements, that allows a larger global coverage, can also be used. An operational algorithm was developed to retrieve AAC scenes from the polarization measurements provided by the POLDER instrument onboard PARASOL (Waquet et al., 2009, 2013a) and was used to provide global analysis of the aerosol above clouds properties (Waquet et al., 2013b). Further, Peers et al., (2015) combined total and polarized radiance measurements to retrieve the aerosol absorption above clouds. A color ratio technic was also developed to retrieve the AAC optical thickness and the corrected cloud optical thickness from total radiance measurements. This method was adapted to OMI UV measurements and MODIS multi-spectral measurements (Torres et al., 2012, Meyer et al., 2015).

And
Concerning Aerosol Above Cloud (AAC), intercomparisons of passive and active retrievals were performed for case studies (Jethva et al., 2013) and for global and multi-year data (Deaconu et al., 2017). The methods developed for passive instruments are however also based on 1D calculations and, so, generally restricted to homogeneous cloudy pixels, for which the 3D effects are minimized. In case of partial cloudy scenes, shadow, cloud enhancement of the clear areas by neighboring clouds can also

modify the retrieved aerosol properties. Errors on the retrieved aerosol properties are in general dependent of the cloud distribution, optical thickness and spatial resolution (Stap et al., 2016a; Stap et al., 2016b).

And added the sentence as:
Concerning AAC retrieval, to our knownledge, no study were conducted to assess errors due to cloud heterogeneity.

The impacts of the 3D effects on the POLDER above cloud AOT operational retrievals in case of fractional cloud were evaluated and presented in Section 5.

**" In this study, the radiative transfer simulations are done at very high spatial resolution, 50 m. although the results are averaged to 7km to "mimic the radiometer measurements and applied the POLDER operational algorithm". Only retrievals at the 7km are presented and analyzed. The reason for the spatial average understandable. But the high-resolution radiative transfer and retrieval results (if any) should also be presented and analyzed for a couple of important reasons. First of all, the 3-D effects are highly dependent on the spatial scale. At small scale (e.g., 50m) the violation of independent pixel approximation (i.e., smoothing, illuminating and shadowing effects) is more important, while at coarser resolution (e.g., 7km) the plane-parallel bias is more important, as pointed out in many previous studies including Zhang et al. (2012). Therefore, the high-resolution results, in combination with the low-resolution results, are very important for us to gain a comprehensive understanding of the problem. Second, the high-resolution results are very relevant to air-borne instruments, such as RSP and HARP. These instruments have been employed in the recent ORACLES field campaign. These air-borne instruments have spatial resolution on the order of 100m. So the results in this paper are highly relevant. Therefore, I strongly suggest the authors add some results and discussion on the high-resolution radiative transfer and retrieval results.**

We know and agree that retrieval results and cloud heterogeneity effects are highly dependent on the spatial resolution. However, as explained in the previous answer, this paper focus on POLDER measurements which are made at a resolution of 6 km x 7 km.  We agree that studies concerning heterogeneity effects are higher spatial resolution would be very valuable. However, we did not make the inversions from the high resolution cloud fields. It is not the scope of the paper and we thing that adding to much information will deserve the whole paper.

**" This paper focuses on the polarimetric remote sensing technique. But it is somewhat disappointing that there is no discussion on the spectral methods for above-cloud aerosol retrievals (e.g., Jethva et al. 2013 and Meyer et al. 2015). As far as I understand, the radiative transfer and retrieval framework used in this study can be easily extended to the spectral method. I'd encourage the authors to take this opportunity to look into the 3-D effects on spectral based above-cloud aerosol retrievals. But I will leave this to the authors to decide whether they will do this in this study or future work. »**

We add the mentioned reference about spectral method in the introduction, see above.

Concerning the method used here to assess cloud heterogeneity effects, for sure, it could be easily extended to above-cloud aerosol retrieval based on spectral method but again we think that is beyond the scope of our paper. It  will maybe be done  a future study (not yet planned).

**" What is not clear from the current paper is how much the retrieval error is due to the 3D effects and how much is due to retrieval algorithm uncertainty.**

You are right, it was not clear in the previous version of the manuscript, we did our best to clarify this point in the new version (see our point by point responses below).

**For example, POLDER has a coarse angular resolution and it seems to me this is partly the reason why the above cloud AOD retrieval error is large in Table 3.**

We do not agree. For the homogeneous cloud considered as infinite, the coarse resolution of POLDER is not an issue. The retrieved AOT from homogeneous cloud input is not significantly different comparing to the AOD input and can be considered as the benchmark value to assess the cloud heterogeneity effects. Retrieved AOT from heterogeneous clouds is then compared to the 1D retrieved AOT. Significant departures are observed for fractional clouds (3D input) in function of the solar zenith angle. As the same radiative transfer model is used for 1D and 3D cases, differences in AOT are then necessarily due to 3D effects that depend on the solar elevation.

To be clearer, we added the following sentence and paragraph in the manuscript :
We remind that the same input AOT is used in the 1D and 3D simulations (AOT of 0.15 at 865 nm).

And further….

As expected, the AOTs retrieved by the algorithm for homogenous clouds (1D input) are close to the input one, whatever the SZA value. The retrieved AOTs only slightly overestimate the input one (0.15) and are respectively equal to 0.18, 0.17, 0.17 for SZA of 20, 40 and 60°. This overestimation is likely due to the approximations used in the retrieval algorithm (e.g. interpolation of the LUTs). Comparing with the retrieved values from homogeneous cloud, significant departures are observed for fractional clouds (3D input) depending on the SZA. The AOTs retrieved at 865 nm are then equal to 0.119, 0.17 and 0.28 for SZA of 20, 40 and 60°, respectively. For a given solar zenith angle, the viewing geometries and the angular resolution are identical for the 1D and 3D inputs. The differences observed in AOT between the 1D and 3D calculations are then necessarily due to 3D effects.

**Also, in the retrieval process based on the Waquet et al. (2013), how much a priori information is given to the retrieval algorithm? Does the retrieval algorithm know, for example, the single scattering albedo of the above-cloud aerosol at each wavelength? In reality, the algorithm certainly does NOT know the aerosol properties. Some discussions are needed to clarify how aerosols are treated in the Waquet et al. (2013) retrieval algorithm and justify the treatment.**

You are right, our description of the aerosol above clouds algorithm was not enough detailed. Actually, Waquet et al. (2013) describes two algorithms: (1) a "research algorithm" that is an optimal estimate method that aims to retrieve a large number of aerosol and cloud parameters and (2) the so-called "operational algorithm" that retrieves the AOT and the Ångström exponent of aerosols above clouds at a global scale. The operational algorithm is the one considered in the present study. This algorithm is based on LUTs' calculations and do not use any a priori information on aerosol and cloud. However, the method uses assumptions on particles microphysics and the LUT is built for a limited set of aerosol and cloud models. The operational algorithm considers six fine mode spherical aerosol models (i.e. effective radius varying between 0.09 and 0.24 microns) and assumes a constant complex refractive index of 1.47+0.01i. The single scattering albedo (SSA) is then also prescribed since this parameter primarily depends on the particles size and on the imaginary part of the complex refractive index (e.g. SSA of 0.91 at 865 nm for mean radius of 0.149 microns and absorption of 0.01). As explained in Peers et al., 2015, polarization measurements are primarily sensitive to scattering processes and mainly provide the scattering AOT. In other words, with polarization measurements at 670 and 865 nm, we retrieve the scattering AOT and with an assumption for the SSA, we provide the total (extinction) AOT. Obviously, the choice of the level of absorption or the choice of the SSA impacts the retrieval of the scattering AOT. Errors due to the assumption made for the complex refractive index were estimated in Peers et al., (2015) and are around 20% for the AOT.

One additional mineral dust model is also considered in this algorithm. One can note that the operational algorithm also uses a specific strategy to retrieve aerosol properties above clouds that depends on the aerosol type (see figure 4 in Waquet et al., 2013).

Finally, a recent global and multi-year comparison between POLDER AOT and CALIOP "depolarization method" AOT retrieved above clouds shows a fairly good agreement (Deaconu et al., 2017). This gives confidence in the operational method developed for POLDER since the depolarization method does not require any assumption in aerosol microphysics to retrieve the AOT.

In the new version of the manuscript, we will provide a better description of the POLDER operational algorithm (i.e. aerosol models, assumptions and retrieval uncertainties, retrieval strategy …) and we will refer to Waquet et al., (2013) for all technical details.

We add the following paragraph in the manuscript :

Waquet et al. (2013) describes two algorithms for Aerosol Above Clouds (AAC) retrieval using POLDER polarization measurements : (i) the research algorithm, that is an optimal estimation method that retrieves a large number of aerosol and cloud parameters, and (ii) the operational algorithm that allows to retrieve the AOT at 865 nm and the Ångström exponent of aerosol above clouds. The "operational algorithm" is the one considered in the present study. This is algorithm is based on LUTs' calculations performed with the successive order of scattering code that assumes a plane-parallel atmosphere (Lenoble et al., 2007). It uses assumptions on particles microphysics : six fine mode spherical aerosol models (i.e. effective radius varying between 0.09 and 0.24 microns) are considered and a constant complex refractive index of 1.47+0.01i is assumed. The errors due to the assumption made for the complex refractive index are around 20% on average for the AOT (Peers et al., 2015). Maximal relative error may reach 25% in case of extreme aerosol events (AOT > 0.6 at 550 nm). One additional non-spherical mineral dust model is also considered in the LUTs.

The operational algorithm uses a specific strategy to retrieve aerosol properties above clouds that depends on the aerosol type and also on the available viewing geometries (see figure 4 in Waquet et al., 2013). In case of fine mode particles, the retrieval is restricted to the use of observations acquired for scattering angles smaller than 130° where polarization measurements are highly sensitive to scattering by fine mode particles (such as biomass burning aerosol) and only weakly sensitive to cloud microphysics. In Figure 6, the dashed line show the increase of the polarized reflectances for scattering angles less than 130° when an aerosol layer is present above a cloud. However, non-spherical particles in the coarse mode such as mineral dust particles, cannot be handled with this method as they do not much polarize light. When dust particles are transported above clouds, they reduce the magnitude of the primary cloud bow. The operational algorithm includes thus the primary bow in order to retrieve the above cloud dust AOT. In this case, as the magnitude of the primary cloud bow primarily depends on the cloud droplet effective radius, it must be estimated or included in the retrieval process. Collocated cloud properties from MODIS at high resolution (1 km × 1 km) are used to characterize and to select the cloudy scenes within a POLDER pixel (6 km × 7 km at nadir) and the MODIS cloud products can then be used in the operational algorithm to estimate the droplets effective radius. As the magnitude of the primary cloud bow is only weakly impacted by the choice of the droplet effective variance, this parameter is assumed to be constant and equal to 0.06. Several filters are eventually applied to obtain a quality-assessed product. For instance, the retrievals are restricted to cloudy pixels associated with cloud optical thicknesses larger than 3.0, since the polarized radiation reflected by the cloud layer is then saturated and does not depend anymore on the cloud optical thickness. Criteria are also used to reject inhomogeneous and fractional cloudy pixels and to avoid cirrus cloud contamination. We refer to Sect. 3.4 in Waquet et al. (2013) for a detailed description of the operational algorithm.

**Related to the last point, the AOD retrieval error could be put into a more meaningful context. For example, what is the relative error in AOD retrieval if the assumption of single-scattering albedo of aerosols is wrong in the retrieval algorithm?**

The relative errors in AOT due to the assumption made for the complex refractive index (1.47-0.01i) were estimated in Peers et al., (2015). Synthetic simulations were generated with different assumptions for the complex refractive index. These synthetic simulations were used as an input to evaluate the algorithm (values of real part of refractive index of 1.42 and 1.54 instead of 1.47 and imaginary part of 0.03 instead of 0.01). The errors on the AOTs are around 20% on average. Maximal relative error may reach 25% in case of an extreme aerosol event (AOT > 0.6 at 550 nm).

[Figure]

*Figure 5 (from peers et al., 2015) Sensitivity of the properties of ACA conditions with different aerosol models. total AOT at 865 nm, COT at 550 nm. Grey lines correspond to the properties of the actual modeled conditions and green lines to those retrieved by the algorithm. The aerosol model of the first column has a refractive index n equal to 1.42 – 0.03i, the second, n = 1.47 – 0.03i and the third, n = 1.52 – 0.03i. Aerosols have an effective radius of 0.1 μm and the effective radius of the cloud water droplets is 10 μm.*

As mentioned in the previous answer, we add this sentence :
Relative errors due to the assumption made for the complex refractive index are around 20% for the AOT, with a maximal error of 25% found in case of aerosols events associated with AOTs larger than 0.6 at 550 nm (Peers et al., 2015).

**How is this error compared with the 3-D effects? Such comparison will help us understand the relative importance of 3-D effects in comparison with some other error sources in the retrieval.**

The errors associated with the retrieval algorithm (i.e. assumptions in the particles microphysics and potential errors introduced by the use of LUTs and interpolation processes were already added in the manuscript:
The retrieved AOTs only slightly overestimate the input one (0.15) and are respectively equal to 0.18, 0.17, 0.17 for SZA of 20, 40 and 60°. This overestimation is likely due to the approximations used in the retrieval algorithm (e.g. interpolation of the LUTs).

and
The errors due to the assumption made for the complex refractive index are around 20% on average for the AOT (Peers et al., 2015). Maximal relative error may reach 25% in case of extreme aerosol events (AOT > 0.6 at 550 nm).

We also completed and rephrased the paragraph discussing the heterogeneity effects on the AAC retrieval.
For a SZA = 20°, the operational algorithm also successfully retrieves the input aerosol model for the homogeneous and fractional cloud. However, the AOT retrieved by the operational algorithm, under the 1D assumption, is underestimated with error between -35 and -40%. For a SZA of 20°, the range of scattering angles effectively used for the retrieval is between 100° and 130°. Polarized reflectances for SZA=20° are not shown but are similar to the ones shown in Figure 7 between 100° and 180°. Over the 100-130°, as shown in Figure 7, 3D polarized reflectances are lower than the 1D ones because of the plane-parallel biases, which explains why the AOT retrieved by the algorithm is

underestimated. However, as the differences are mainly due the plane-parallel bias, which is similar for the two wavelengths, the cloud heterogeneity effects do not affect the selection of the best aerosol model.

For SZA = 60°, the range of scattering angles used is between 60° and 130°. Between 60° and 90°, there is an increase of the forward scattering signal due to 3D effects, which is interpreted by the operational algorithm as an increase in the AOT. We note also that 3D effects bias the aerosol model for this case as a smaller value of Ångström exponent (corresponding to a larger effective radius) is retrieved for the fractional cloud. The retrieved AOT is thus higher (AOT of 0.28 comparing to 0.17) with a relative error up to 65%. For SZA=60°, the 3D effects consist in an increase of the polarized signal because of additional scattering in the clear sky parts. This increase is higher at 865 nm than at 670 nm. This leads to the selection by the algorithm of an erroneous model with a smaller Angström exponent.

Deaconu, L.T., Waquet, F., Josset, D., Ferlay, N., Peers, F., Thieuleux, F., Ducos, F., Pascal, N., Tanré, D., Pelon, J., Goloub, P., 2017. Consistency of aerosols above clouds characterization from A-Train active and passive measurements. Atmos Meas Tech 10, 3499–3523. https://doi.org/10.5194/amt-10-3499-2017

Lenoble, J., Herman, M., Deuzé, J.L., Lafrance, B., Santer, R., Tanré, D., 2007. A successive order of scattering code for solving the vector equation of transfer in the earth's atmosphere with aerosols. J. Quant. Spectrosc. Radiat. Transf. 107, 479–507. https://doi.org/10.1016/j.jqsrt.2007.03.010

Peers, F., Waquet, F., Cornet, C., Dubuisson, P., Ducos, F., Goloub, P., Szczap, F., Tanré, D., Thieuleux, F., 2015. Absorption of aerosols above clouds from POLDER/PARASOL measurements and estimation of their direct radiative effect. Atmos Chem Phys 15, 4179–4196. https://doi.org/10.5194/acp-15-4179-2015

Stap F. A., Hasekamp O. P., Emde C., Röckmann T., 2016. Multiangle photopolarimetric aerosol retrievals in the vicinity of clouds: Synthetic study based on a large eddy simulation. J. Geophys. Res. Atmospheres 121, 12,914-12,935. https://doi.org/10.1002/2016JD024787

Stap, F.A., Hasekamp, O.P., Emde, C., Röckmann, T., 2016. Influence of 3D effects on 1D aerosol retrievals in synthetic, partially clouded scenes. J. Quant. Spectrosc. Radiat. Transf. 170, 54–68. https://doi.org/10.1016/j.jqsrt.2015.10.008

Waquet, F., Cornet, C., Deuzé, J.-L., Dubovik, O., Ducos, F., Goloub, P., Herman, M., Lapyonok, T., Labonnote, L.C., Riedi, J., Tanré, D., Thieuleux, F., Vanbauce, C., 2013. Retrieval of aerosol microphysical and optical properties above liquid clouds from POLDER/PARASOL polarization measurements. Atmos Meas Tech 6, 991–1016. https://doi.org/10.5194/amt-6-991-2013

---

## Author Comment (AC4) · 2 Jun 2018

**Reply to F. Xu**

The authors would like to thank F Xu for its valuable comments and suggestions that allow to improve the paper.

Please, find below the answers to the comments

1) **The authors may double check Eq.(1) as it is more like a definition for bidirectional reflectance factors (BRF) instead of for "total reflectance"? In addition, to define polarized reflectance, it is better to use sqrt(Q^2+U^2+V^2) instead of "I" in Eq(1) for clarity.**

Right, we modified the definitions and wrote page 7;

From these 3D cloud fields, we simulated the total and polarized bidirectional reflectances function for the viewing zenith angle θ and the viewing azimuthal angle φ. By convenience, in the following, we call them total reflectance R and polarized reflectance Rp:

$$R(\theta, \varphi) = \frac{\pi . I(\theta, \varphi)}{F_0 cos\theta_0}$$

$$R_p(\theta, \varphi) = \frac{\pi}{cos\theta_0} \sqrt{Q^2(\theta, \varphi) + U^2(\theta, \varphi) + V^2(\theta, \varphi)}$$

where $I(\theta, \varphi), Q(\theta, \varphi), U(\theta, \varphi)$ and $V(\theta, \varphi)$ are the four Stokes parameters in $W.m^{-2}.sr^{-1}$, $F_0$ the solar flux in W.m-2 and $\theta_0$ the solar zenith angle.


2) **Does the AOT retrieval closure test use the simulated signals from the whole scattering angular range from 60 to 180 degree ? It can be observed from Figs. 4 and 6 that the 3D impact is more remarkable in the scattering angular ranges from 60 to 80 degrees and from 160 to 180 degrees. What if the authors try doing the aerosol optical thickness (AOT) retrieval using the signals from 80-160 degrees range only (where 1D RT apparently has less plane-parallel bias) and re-evaluating the 3D impact on AOT retrieval ? I assume the aerosol information residing in this reduced angular range may be good enough for AOT retrieval (and may result in reduced error).**

3D effects are clearly visible for forward scattering geometries (i.e. scattering angle ranging between 60 and 80°) in case of low solar elevation (see figure 6). The scattering angle range sampled between 60-80° is not necessarily useful for an accurate retrieval of the above cloud AOT. So, reducing the use of forward scattering geometries restricted to scattering angle values larger than 60° will help in reducing retrieval errors in AOT. We will include it in the paper, thanks. However, other large errors due, to 3D effects, are also observed in the primary bow (around 140°) in case of fractional clouds that can be neglected.

We added in the conclusion section :
These results mainly show that 3D effects for fractional clouds are primarily significant at forward scattering geometries in case of low solar elevation (scattering angle < 80° and SZA of 60°) and in the rainbow region (scattering angle of about 140° +/- 5°). The range of scattering angles sampled between 60 and 80° is not necessarily useful for an accurate retrieval of the above cloud AOT. So, reducing the range of scattering angles to scattering angle values larger than 80° will help to reduce the errors associated with the AOT retrievals. The algorithm largely overestimates the AOT when the primary bow is included in the retrieval process and when forward and side scattering viewing geometries are not available. This result suggests that polarized measurements acquired for this configuration should not be used for AAC properties retrievals, at least with a retrieval algorithm based on 1D calculations.

**3) For solar incidence angles 20 and 40 degrees, the cloudbow signals (e.g. in the principal plane) should appear in two sides around incidence ray. And their magnitudes should be somehow different. But such a difference is not observed in Figs. 4-6. Is this due to the signals at the same scattering angles are just averaged regardless of the difference in viewing angles ? It may be more clear if the authors plot both of them in those figures.**

In Figures 4-6, (now Figures 5 to 7), we plotted only the figures for the case SZA=60° which allows to display all the scattering angular range (between 60 and 180°). Figure RC4-1 shows polarized reflectances for three solar incidence angles (left) and absolute difference between 1D and 3D polarized reflectances (right). For a same range of scattering angles, the effects of cloud heterogeneity are very similar for all the solar incidence angles, so we chose to plot only the graphs for SZA=60°. Note for SZA=20° and 40°, the two branches appear representing the two sides of the scattering angles around 180°.

We added it in the manuscript at the end of section 4-1:
Figures5 illustrate results obtained for simulations for SZA=60° with a scattering angular range between 60 and 180°. Note that for SZA = 20° and SZA = 40°, the plots are similar with a reduced scattering angular range that is between 100° and 180° for SZA=20° and between 80° and 180° for SZA=40°. Consequently, for SZA = 20 ° and SZA=40 ° the attenuation due to the plane-parallel bias is the main impact of the measurements.

And removed the sentence "3D cloud radiative effects are thus important, particularly in the forward direction, but it is important to note that such 3D effects are weaker for smaller SZA and almost not present for SZA=20°." that speak about shadowing effects but may be confusing.

[Figure]

*Figure RC4-1: (left) Polarized reflectances at 865nm as a function of scattering angle for three solar zenith angles. Dashed lines are for homogeneous cloud and solid for heterogenous cloud. The case presented is the one with a biomass burning aerosol layer above. (Right) Absolute differences between 3D and 1D polarized reflectances.*

**4) It may be necessary to describe a little more on the criterion for setting 50 m as the small scale (pixel scale). Is this set up due to the sufficiency in ensuring a) representativeness of cloud microphysical property variation and/or b) accuracy of cloud signals in a certain scale ?**

The choice of pixel scale dimension is not a simple question. It depends on the studied scale, here it is a POLDER pixel that is 7 km x 7 km. We assumed that the description of sub-pixel variability at 50m is sufficient to simulate correctly POLDER observation. In addition, interaction between cloud and radiation can be estimated using the mean free path of the photon. For an homogeneous cloud, it is computed as the inverse of the extinction coefficient $\sigma$: $MFP = \frac{1}{\sigma} = \frac{H}{COT}$ where H is the geometrical thickness and COT the cloud optical thickness of the cloud. In the paper the mean COT is 10 and H≈700m, consequently the $MFP \approx 70m$ so above the 50m resolution and it is even larger in heterogeneous media (Davis and Marshak, 2004). Availability of computer memory and time computation were also considered for this choice .

We add:
We assumed that the description of the cloud fields at 50m is sufficient to simulate correctly the POLDER observation at 7 km x 7 km. Moreover, the interaction between cloud and radiation can be characterized by the mean free path (MFP) of the photon that is of the order of 70 m ($MFP = \frac{1}{\sigma} = \frac{H}{COT}$) for the equivalent homogeneous cloud and larger for heterogeneous cloud (Davis and Marshak, 2004). Availability of computer memory and time computations were also considered.

---

## Author Comment (AC5) · 2 Jun 2018

**Cloud heterogeneity on cloud and aerosol above cloud properties retrieved from simulated total and polarized reflectances**

Céline Cornet1, Laurent C-Labonnote1, Fabien Waquet1, Frédéric Szczap2, Lucia Deaconu1, Frédéric 5 Parol1, Claudine Vanbauce1, François Thieuleux1, Jérôme Riédi1

[revised manuscript text omitted]

|-----------------------|--|

2017). Concerning aerosol above cloud retrieval, to our knownledge, no study were conducted to assess errors due to cloud heterogeneity.

[revised manuscript text omitted]

inverse of the extinction coefficient so to about 70m) but also considering computation time and virtual memory availability.

The three generated clouds have the same mean optical thickness close to 10 at 865 nm. We created two stratocumulus clouds and one cumulus cloud. The latter is the result of instabilities of the boundary

- 5 layer and lead to fractional cloud cover and larger heterogeneity parameter (Kawai and Teixeira, 2011). The flat and bumpy clouds representing overcast stratocumulus clouds have, the same heterogeneity parameter across the 140x140 pixels,  $\rho = 0.6$ , which is a typical value for stratocumulus cloud. The cumulus cloud has a fractional cloud cover equal to 0.76 and a heterogeneity parameter equal to 1.12 setting clear sky pixels to null values (0.95 if computed only with the cloudy pixels). These values are
- 10 typical values obtained from Landsat data (Barker et al., 1996) for stratocumulus and cumulus clouds. Figure 1 shows the vertical profiles of potential temperature and of vapor mixing ratio prescribed in this study to generate the three cloud fields. Globally, the vertical profiles of potential temperature and vapor mixing ratio give the cloud position. The mean cloud top height is mainly determined by the height where the potential temperature increases and the vapor mixing ratio decreases. Cloud top height
- 15 fluctuations (shapes of top bumps) are mainly the result of the intensity of the vertical gradient of the potential temperature and vapor mixing ratio.

Figure 2 shows the horizontal cloud optical thickness field and a vertical profile through each cloud. In this study, we focus on the effects of the optical thickness heterogeneity, which is supposed, in real

20 cloud, to be more important than the microphysical heterogeneity (Magaritz-Ronen et al., 2016). Consequently, the cloud droplet size distribution is assumed uniform everywhere in the cloud and follows a log-normal distribution with an effective radius of 11 µm and an effective variance of 0.02.

From these 3D cloud fields, we simulated the total and polarized bidirectional reflectances function for
the viewing zenith angle θ and the viewing azimuthal angle φ. By convenience, in the following, we call them total reflectance *R* and polarized reflectance *Rp*:

| over the mean optical thickness)                                                                                                       |
|----------------------------------------------------------------------------------------------------------------------------------------|
|                                                                                                                                        |

$$R(\theta, \varphi) = \frac{\pi . I(\theta, \varphi)}{F_0 cos \theta_0}$$
$$R_p(\theta, \varphi) = \frac{\pi}{F_0 cos \theta_0} \sqrt{Q^2(\theta, \varphi) + U^2(\theta, \varphi) + V^2(\theta, \varphi)}$$

where  $I(\theta, \varphi), Q(\theta, \varphi), U(\theta, \varphi)$  and  $V(\theta, \varphi)$  are the four Stokes parameters in W.m-2.sr-1,  $F_{0}$  the solar 5 flux in W.m-2 and  $\theta_{0}$  the solar zenith angle.

Reflectances for three solar incidence angles 20°, 40° and 60° are computed with the 3D radiative transfer model, 3DMCPOL It is a forward Monte-Carlo model able to compute radiative reflected or transmitted Stokes vector as well as upwelling and downwelling fluxes in three-dimensional

- 10 atmospheres. Initially, developed for solar radiation (Cornet et al., 2010), it was next extended to thermal radiation (Fauchez et al., 2014). To save time and for an accurate computation of reflectances, the local estimate method (Marshak and Davis, 2005) is used. Periodical boundary conditions at the horizontal domain limits are used. For highly peaked phase function, the potter truncation is implemented. Molecular scattering is computed according to the pressure profile. A heterogeneous
- 15 surface can also be specified with Lambertian reflection, ocean or snow bidirectional function. The model participated and was improved during the Intercomparison of Polarized Radiative Transfer model (IPRT) on homogeneous cloud cases (Emde et al., 2015) and on 3D cloud cases (Emde et al., 2018). Simulations are run with a total of 107 photons and 109 photons for the homogeneous and heterogeneous clouds respectively. The Monte-Carlo uncertainties are estimated with the computation of standard
- 20 deviation with 10 and 50 independent realizations of 106 and 20.106 photons for the homogeneous and heterogeneous cloud respectively. For the homogeneous case, the relative standard deviation is below 0.12% for the total reflectances and below 1.2% for the polarized reflectances. For the heterogeneous clouds, at 50m resolution, the mean relative standard deviation is below 1.3% for the total reflectances. For polarized reflectances at 50 m, the mean relative standard deviation varies according to the angular
- 25 geometry and is between 2% and 107% for very small reflectance values with a mean value of 23%. At 7km, as the reflectances are averaged, relative standard deviation values are much lower below 0.01% and 0.8% for total and polarized reflectances respectively.

| Mis en | forme | : | Exposant |
|--------|-------|---|----------|
|--------|-------|---|----------|

where isare the radiance in W.m-2.sSr-1,  $F_0$  the solar flux in W.m-2 and  $\theta_0$  the zenith solar incidence angle.

| Mis en | forme : Non Surlignage |
|--------|------------------------|
| Mis en | forme : Non Surlignage |
| Mis en | forme : Exposant       |
| Mis en | forme : Non Surlignage |
| Mis en | forme : Non Surlignage |
| Mis en | forme : Non Surlignage |

At this stage, molecular scattering is integrated but no aerosols. To remain consistent with assumptions made within POLDER operational algorithm, an oceanic surface with a wind speed of 7 m.s-1 is included for total reflectances while a black surface is included for polarized reflectances. Indeed, for 5 retrieval using polarized reflectances, the multi-angular ability of POLDER provides the advantage of

not using the directions close to the sun-glint where the polarized reflectances can be high.

As POLDER measures up to 16 directions, we simulate reflectances for 16 POLDER typical zenith observation angles in the solar plane. Total reflectances of the three clouds are presented in Figure 3 (first column) with a 50 m spatial resolution for a solar incidence angle of 60° in the cloudbow direction (40° from the backward direction). Polarized reflectance, fields are discussed in Section 4.1.

**3 Impacts on total reflectances and consequences for optical thickness and albedo retrieval**

10

We averaged spatially the 50 m resolution reflectances fields at 7 km x 7 km to mimic the radiometer measurements and applied the POLDER operational algorithm on these synthetic measurements to

- 15 obtain cloud optical thickness and albedo. In order to assess the retrieval errors due to the cloud homogeneous assumption without biases due to differences in reflectance computations, we also computed the 1D reflectances of the three equivalent homogeneous clouds, which are subsequently used for retrieval to act as references for the inhomogeneous cloud retrievals. The COT of the equivalent homogeneous clouds is the mean COT of the heterogeneous clouds, and their cloud top and base
- 20 altitudes correspond to the maximum and minimum altitudes of the respective homogenous clouds. The mean optical thickness, and the cloud top and base altitudes corresponding to the maximal and minimal altitudes of the heterogeneous clouds are used.

Figure 4 summarizes the results obtained for the retrieved cloud optical thickness for the three solar zenith angles and the four cases, namely the homogeneous (1D), the flat, the bumpy and the fractional

25 cloud, The optical thicknesses are plotted as a function of sensor zenith angle with negative value corresponding to backward scattering directions and positive value to forward scattering directions. The homogeneous cloud values (1D) are only plotted for control and we observe logically that the retrieved

| r | 1 | ١ |  |
|---|---|---|--|
|   |   |   |  |
|   |   |   |  |
|   |   |   |  |

| Mis en | forme : | Exposant |
|--------|---------|----------|
|--------|---------|----------|

|-----------|--|

| Supprime: The choice of not considering the surface for algorithm
using polarized reflectances rely on the fact that polarized
reflectances are not affected by surface for cloud optical thickness
larger than 4. |   |
|-----------------------------------------------------------------------------------------------------------------------------------------------------------------------------------------------------------------------------|---|

|---------|-------------------------------------------------------------------|
| ······( | Supprimé: 3                                                       |

**Supprime:** Note that in the three cases, the operational algorithm retrieves a cloud cover equal to one.

value is almost constant and close to 10, independent of the solar incidence angle, since the same assumption (1D homogeneous cloud) is used in both the forward simulation and retrieval algorithm. Slight differences appear because of inclusion of aerosol optical thickness in the forward model used to build the look-up table (Buriez et al., 1997) but not in our simulations. The small angular difference in 5 the backward direction at 20° can be attributed to interpolation in the LUT.

- Looking at results concerning the heterogeneous clouds (3D), we clearly note, in the angular range between about -30° and +30°, the plane-parallel bias, which leads to retrieve optical thicknesses lower than the mean optical thickness. At nadir view, the relative error is between -10 and -20% both for the flat and bumpy cloud and is much larger for the fractional cloud, between -35 and -50%. The flat and
- 10 bumpy clouds were built with the same heterogeneity parameter ( $\rho$ =0.6) whereas the fractional cloud has a larger heterogeneity parameter including the zeros ( $\rho$ =1.12) due to its fractional nature. That confirms that heterogeneity parameters can be at first order used to characterize plan-parallel bias (Cahalan et al. 1994, Szczap et al., 2000a).

For solar zenith angle (SZA) equal to 20°, the retrieved optical thickness is almost independent of the 15 observation geometry whatever the cloud type, while for SZA=60°, significant differences between view, angles are observed. We note indeed a strong decrease of the retrieved optical thickness value in the forward scattering direction leading to a relative bias on the retrieved optical thickness between -40% for the flat and bumpy cloud and -70% for the fractional cloud. On the contrary, we can notice an increase of the retrieved optical thickness value in the backscatter direction (relative bias ranging from

- 20 +3% for the flat cloud, +43% for the bumpy cloud and +21% for the fractional cloud). This angular behavior was already simulated by several authors at the resolution of 1 km (Loeb et al., 1998; Varnai, 2000; Iwabuchi and Hayasaka, 2002; Zinner and Mayer, 2006) and agrees with POLDER observations (Buriez et al., 2001; Zeng et al., 2012). In the backscatter directions, the cloud sides illuminated by the Sun make, the cloud brighter, in contrast to the forward direction where, cloud sides are in the shadow
- 25 (Varnai and Davies, 1999). These effects are visible for the bumpy cloud but are much less pronounced for the flat cloud. The heterogeneity parameter thus seems well adapted to characterize quantitatively the plane-parallel bias (Szczap et al., 2000a) but not sufficient to characterize the amplitude of the 3D effects. Indeed, the flat and bumpy clouds, which are characterized by the same heterogeneity

[revised manuscript text omitted]

of pixels exhibits polarized reflectance higher than the maximum value predicted by the 1D homogeneous cloud model (yellow pixels) and thus cannot be obtained with 1D radiative transfer simulation : at 490 nm, their ratio reaches 41% of the total number of pixels for the flat cloud, 52% for the bumpy cloud and 38% for the fractional cloud. This phenomenon of illumination and shadowing was already highlighted with simply a step cloud in Cornet et al. (2010).

In the forward direction ( $\Theta$ =60°) at 490 nm (third column in Figure 3), the "shadow areas" are not dark anymore contrarily to the total reflectance (first column in Figure 3) and appear even brighter than cloudy part. For short wavelength and forward scattering angles, molecular signal is stronger than the cloud signal and thus enhances the polarized signal in the shadow parts.

5

- 10 In Figure  $\underline{5}_{\phi}$  we plot the average polarized reflectances as would be measured by POLDER at 7 kmx 7 km resolution as a function of the scattering angle  $\Theta$  for a solar zenith angle SZA=60°, and for the three wavelengths. As we can see in Figure  $\underline{5}_{\varphi}$ , the main differences between homogeneous and heterogeneous clouds appear in the cloudbow direction ( $\Theta$ =140°) and in the forward direction ( $\Theta$  < 80°). In the cloudbow direction, the 3D polarized reflectances are lower than the 1D ones for the three
- 15 clouds. Similar to the total reflectances, this is mainly due to the plane-parallel bias. In these directions, the relative differences (Figure 5b) are about -9%, -12% and -35% for the flat, bumpy and fractional cloudx respectively. We note that the relative difference is slightly lower for 490 nm because of the smoothing effects by molecular scattering above the cloud.

In the forward scattering direction, the consequences of the 3D effects in terms of absolute polarized 20 reflectances appear differently depending on the wavelength. At 490 nm, the 3D effects enhance the

- absolute polarization, while at 865 nm they reduce it. At 490 nm, atmospheric molecular scattering is very strong. The 3D polarized reflectances appear greater than the 1D ones because, as seen in Figure  $\underline{3}_{acc}$ the polarization in the shadow parts of the cloud is enhanced by this molecular scattering. At 865 nm, the shadow parts appear dark with small positive values that reduce the negative polarization of the
- 25 cloud and consequently the absolute polarization. The relative difference (Figure 5b) is consequently positive for 490 nm (about +55% for the fractional cloud) and negative for 865 nm (about -75% for the fractional cloud). At 670 nm, the polarized reflectance in the shadow part is only slightly enhanced by the molecular scattering but more compared to 865 nm. Polarized reflectances thus become positive for

|-----------|----------|
| Supprimé: | pictures |
| Supprimé: | As for   |

| supprime: according to                                 |  |
|--------------------------------------------------------|--|
|                                                        |  |
|                                                        |  |
|                                                        |  |
| n                                                      |  |
| Supprime: 3                                            |  |
|                                                        |  |
|                                                        |  |
|                                                        |  |
|                                                        |  |

the fractional cloud but not for the flat and bumpy clouds. Note that in the backward direction, the polarized reflectances are very weak so no heterogeneity or 3D effects can be detected.

Figures5 illustrate results obtained for simulations for SZA=60° with a scattering angular range between 60 and 180°. Note that for SZA = 20° and SZA = 40°, the plots are similar with a reduced scattering 5 angular range that is between 100° and 180° for SZA=20° and between 80° and 180° for SZA=40°,

Consequently, for SZA =  $20^{\circ}$  and SZA= $40^{\circ}$  the attenuation due to the plane-parallel bias is the main impact of the measurements.

**4.2 Consequences for droplet size distribution and cloud top pressure retrievals**

- 10 The polarized signal is used as input of a POLDER retrieval algorithm developed to retrieve effective radius, effective variance and cloud top pressure. It uses the polarized information as presented in Bréon and Goloub (1998). The position of the cloudbow as well as the position of the supernumerary bows gives information on the effective radius. The amplitude of the supernumerary bows gives information on the effective variance of the cloud droplet size distribution. For cloud top pressure, the algorithm
- 15 uses the information given by the molecular scattering which depends, in the forward scattering directions, on the atmospheric air mass factor (Goloub et al., 1994). The algorithm, under implementation in the POLDER operational algorithm, is based on an optimal estimation method (Rodgers, 2000) and provides errors associated to each of the retrieved parameters. It is also possible to add in the forward model variance-covariance matrix an error due to the non-retrieved parameter.
- 20 Following previous computations made in (Waquet et al., 2013a), for the misrepresentation of 3D effects, the error added in the variance-covariance matrix on the reflectances is 7.5% in the directions close to the cloudbow and 5% elsewhere.

The retrieved values obtained with this algorithm based on the homogeneous cloud assumption, are presented in Table 2, We use again the homogeneous cloud (1D cloud) to check the consistency of our simulations. For all clouds, even if differences in polarized reflectances are large in amplitude, the retrieval algorithm still capture the general angular features of the three wavelengths, which results of

|--------------------------------------------------------------------------------------------------------------------------------------------------------------------------------------------------------------------------|--|
|                                                                                                                                                                                                                          |  |
| Supprime: 4                                                                                                                                                                                                              |  |
particularly in the forward direction, but it is important to note that
such 3D effects are weaker for smaller SZA and almost not present
for SZA=20°. |  |

|-----------|---|
|           |   |

|-----------|---------------|
| Supprimé: | error         |
| Supprimé: | direction     |
| Supprimé: | As previously |
| Supprimé: | quite         |

[revised manuscript text omitted]

Non Italique, Anglais (E.U.) |
|---------------------------------------------------------------------------------------------|
Non Italique, Anglais (E.U.) |
Non Italique, Anglais (E.U.) |
Non Italique, Anglais (E.U.) |

Non Italique, Anglais (E.U.) |
|---------------------------------------------------------------------------------------------|
Non Italique, Anglais (E.U.) |
Non Italique, Anglais (E.U.) |
Non Italique, Anglais (E.U.) |
Non Italique, Anglais (E.U.) |
Non Italique, Anglais (E.U.) |
Non Italique, Anglais (E.U.) |
Non Italique, Anglais (E.U.) |
Italique, Anglais (E.U.)              |
Italique, Anglais (E.U.)              |
Italique, Anglais (E.U.)              |
Non Italique                 |
Non Italique                 |
Non Italique, Anglais (E.U.) |
Non Italique. Anglais (E.U.) |
| (                                                                                           |

cloudy scenes within a POLDER pixel ( $6 \text{ km} \times 7 \text{ km}$  at nadir) and the MODIS cloud products can then be used in the operational algorithm to estimate the droplets effective radius. As the magnitude of the primary cloud bow is only weakly impacted by the choice of the droplet effective variance, this parameter is assumed to be constant and equal to 0.06. Several filters are eventually applied to obtain a

5 quality-assessed product. For instance, the retrievals are restricted to cloudy pixels associated with cloud optical thicknesses larger than 3.0, since the polarized radiation reflected by the cloud layer is then saturated and does not depend anymore on the cloud optical thickness. Criteria are also used to reject inhomogeneous and fractional cloudy pixels and to avoid cirrus cloud contamination. We refer to Sect. 3.4 in Waquet et al. (2013) for a detailed description of the operational algorithm.

10

In the POLDER operational algorithm, the underneath cloud is assumed to be homogeneous. Empirical criterions are used to reject heterogeneous and fractional cloudy pixels but a misclassification of the cloudy scenes is still possible. Moreover, it is also important to evaluate the AOT retrieval errors due to 3D effects in case of fractional cloud covers. These scenes, for which aerosols and clouds are

- 15 potentially mixed, remain untreated and are of primarily importance for climate studies. In the following, we investigate the possibility to use the operational algorithm to treat these scenes and we evaluate the biases observed in the polarized reflectances and in the AOT retrieval errors due to 3D effects. In order to check the AOT value retrieved for such cases, we use the 3D polarized reflectances generated for the fractional cloud case, with and without aerosol, and we used these 3D simulations as
- 20 inputs for the operational algorithm. Note, that for the synthetic retrievals discussed here below, we assumed that the operational algorithm knows the effective radius and effective variance of the cloud droplets.

The 3D polarized reflectances used as input of the algorithm and the ones simulated after the adjustment of the aerosol model and optical thickness are plotted in Figure 7 (solid lines). When a large scattering 325 angular range is available (between 60 and 180°), the algorithm works in an efficient way. The lateral

polarized reflectances in scattering angular range between 80° and 120° exhibit low or negative values. Consequently no aerosol (AOT=0) were retrieved. We note however that the primary cloudbow is not

Non Italique, Anglais (E.U.)                                                                                                                                                                                                                                                                                                                                                                                                                                                                                                                                                                                                                                                                                                                                                                                                                                                 |
|-------------------|---------------------------------------------------------------------------------------------------------------------------------------------------------------------------------------------------------------------------------------------------------------------------------------------------------------------------------------------------------------------------------------------------------------------------------------------------------------------------------------------------------------------------------------------------------------------------------------------------------------------------------------------------------------------------------------------------------------------------------------------------------------------------------------------------------------------------------------------------------------------------------------------------------------------------------------------|
Non Italique, Anglais (E.U.)                                                                                                                                                                                                                                                                                                                                                                                                                                                                                                                                                                                                                                                                                                                                                                                                                                                 |
Non Italique, Anglais (E.U.)                                                                                                                                                                                                                                                                                                                                                                                                                                                                                                                                                                                                                                                                                                                                                                                                                                                 |
Non Italique                                                                                                                                                                                                                                                                                                                                                                                                                                                                                                                                                                                                                                                                                                                                                                                                                                                                 |
Non Italique, Anglais (E.U.)                                                                                                                                                                                                                                                                                                                                                                                                                                                                                                                                                                                                                                                                                                                                                                                                                                                 |
Non Italique                                                                                                                                                                                                                                                                                                                                                                                                                                                                                                                                                                                                                                                                                                                                                                                                                                                                 |
Non Italique, Anglais (E.U.)                                                                                                                                                                                                                                                                                                                                                                                                                                                                                                                                                                                                                                                                                                                                                                                                                                                 |
Italique, Anglais (E.U.)                                                                                                                                                                                                                                                                                                                                                                                                                                                                                                                                                                                                                                                                                                                                                                                                                                                              |
Après : 12 pt                                                                                                                                                                                                                                                                                                                                                                                                                                                                                                                                                                                                                                                                                                                                                                                                                                                                  |
|                   |                                                                                                                                                                                                                                                                                                                                                                                                                                                                                                                                                                                                                                                                                                                                                                                                                                                                                                                                             |
on the difference between Rayleigh pressure based on the use of
polarized reflectance values due to molecular scattering above the
cloudthat is enhanced by aerosol scattering.(Goloub et al., 1994) and
oxygen cloud top pressure which used differential absorption
measurement in the oxygen A-band (Vanbauce et al., 1998).
(Waquet et al., 2009). (Waquet et al., 2009). The AOT above cloud is
next retrieved using the fast algorithm of (Waquet et al., 2013).
Information on AOT is given by the cloudbow attenuation near 140°
and the increase of polarized signal in the forward scattering directing                                                                                                                                                                                   |
on the difference between Rayleigh pressure based on the use of
polarized reflectance values due to molecular scattering above the
cloudthat is enhanced by aerosol scattering,(Goloub et al., 1994) and
oxygen cloud top pressure which used differential absorption
measurement in the oxygen A-band (Vanbauce et al., 1998).
(Waquet et al., 2009). (Waquet et al., 2009). The AOT above cloud is
next retrieved using the fast algorithm of (Waquet et al., 2013).
Information on AOT is given by the cloudbow attenuation near 140°
and the increase of polarized signal in the forward scattering direction
Déplacé vers le bas [2]: (Waquet et al., 2009). (Waquet et al.,                                                                                                                |
on the difference between Rayleigh pressure based on the use of
polarized reflectance values due to molecular scattering above the
cloudthat is enhanced by acrosol scattering.(Goloub et al., 1994) and
oxygen cloud top pressure which used differential absorption
measurement in the oxygen A-band (Vanbauce et al., 1998).
(Waquet et al., 2009). (Waquet et al., 2009). The AOT above cloud is
next retrieved using the fast algorithm of (Waquet et al., 2013).
Information on AOT is given by the cloudbow attenuation near 140°
and the increase of polarized signal in the forward scattering direction
Déplacé vers le bas [2]: (Waquet et al., 2009). (Waquet et al.,
Déplacé (insertion) [2]                                                                                     |
on the difference between Rayleigh pressure based on the use of
polarized reflectance values due to molecular scattering above the
cloudthat is enhanced by acrosol scattering (Goloub et al., 1994) and
oxygen cloud top pressure which used differential absorption
measurement in the oxygen A-band (Vanbauce et al., 1998).
(Waquet et al., 2009). (Waquet et al., 2009). The AOT above cloud is
next retrieved using the fast algorithm of (Waquet et al., 2013).
Information on AOT is given by the cloudbow attenuation near 140°
and the increase of polarized signal in the forward scattering direction
Déplacé vers le bas [2]: (Waquet et al., 2009). (Waquet et al.,
Déplacé (insertion) [2]
on the difference between Rayleigh pressure based on the use of
polarized reflectance values due to molecular scattering above the
cloudthat is enhanced by aerosol scattering (Goloub et al., 1994) and
oxygen cloud top pressure which used differential absorption
measurement in the oxygen A-band (Vanbauce et al., 1998).
(Waquet et al., 2009). (Waquet et al., 2009). The AOT above cloud is
next retrieved using the fast algorithm of (Waquet et al., 2013).
Information on AOT is given by the cloudbow attenuation near 140°
and the increase of polarized signal in the forward scattering direction
Déplacé vers le bas [2]: (Waquet et al., 2009). (Waquet et al.,
Déplacé (insertion) [2]
on the difference between Rayleigh pressure based on the use of
polarized reflectance values due to molecular scattering above the
cloudthat is enhanced by aerosol scattering.(Goloub et al., 1994) and
oxygen cloud top pressure which used differential absorption
measurement in the oxygen A-band (Vanbauce et al., 1998).
(Waquet et al., 2009). (Waquet et al., 2009). The AOT above cloud is
next retrieved using the fast algorithm of (Waquet et al., 2013).
Information on AOT is given by the cloudbow attenuation near 140°
and the increase of polarized signal in the forward scattering direction
Déplacé vers le bas [2]: (Waquet et al., 2009). (Waquet et al.,
Déplacé (insertion) [2]
on the difference between Rayleigh pressure based on the use of
polarized reflectance values due to molecular scattering above the
cloudthat is enhanced by aerosol scattering.(Goloub et al., 1994) and
oxygen cloud top pressure which used differential absorption
measurement in the oxygen A-band (Vanbauce et al., 1998).
(Waquet et al., 2009). (Waquet et al., 2009). The AOT above cloud is
next retrieved using the fast algorithm of (Waquet et al., 2013).
Information on AOT is given by the cloudbow attenuation near 140°
and the increase of polarized signal in the forward scattering direction
Déplacé vers le bas [2] : (Waquet et al., 2009). (Waquet et al.,
Déplacé (insertion) [2]
on the difference between Rayleigh pressure based on the use of
polarized reflectance values due to molecular scattering above the
cloudthat is enhanced by aerosol scattering.(Goloub et al., 1994) and
oxygen cloud top pressure which used differential absorption
measurement in the oxygen A-band (Vanbauce et al., 1998).
(Waquet et al., 2009). (Waquet et al., 2009). The AOT above cloud is
next retrieved using the fast algorithm of (Waquet et al., 2013).
Information on AOT is given by the cloudbow attenuation near 140°
and the increase of polarized signal in the forward scattering direction
Déplacé vers le bas [2]: (Waquet et al., 2009). (Waquet et al.,
Déplacé (insertion) [2]
on the difference between Rayleigh pressure based on the use of
polarized reflectance values due to molecular scattering above the
cloudthat is enhanced by aerosol scattering.(Goloub et al., 1994) and
oxygen cloud top pressure which used differential absorption
measurement in the oxygen A-band (Vanbauce et al., 1998).
(Waquet et al., 2009). (Waquet et al., 2009). The AOT above cloud is
next retrieved using the fast algorithm of (Waquet et al., 2013).
Information on AOT is given by the cloudbow attenuation near 140°
and the increase of polarized signal in the forward scattering directing
Déplacé vers le bas [2]: (Waquet et al., 2009). (Waquet et al.,
Déplacé (insertion) [2]

Non Italique, Anglais (E.U.)

well modelled by the 1D simulation provided by the operational algorithm. In the POLDER measurements, the range of sampled scattering angles varies, with the geographical position. In some cases, the scattering angle range sampled by the instrument can be quite narrow. We tested the algorithm without observations acquired for scattering angles smaller than 120°, (dashed lines in Figure

5 7). The cloudbow signal is then better matched but the inversion method retrieves erroneous AOT values of 0.31 at 670 nm and 0.28 at 865 nm instead of zero for both.

A second test is made with simulated reflectances including, a biomass-burning aerosol layer lofted above the fractional cloud. For the simulation, the AOT of the aerosol layer is fixed to 0.28 and 0.15, the single scattering albedo to 0.93 and 0.91 at 670 and 865 nm respectively. In order to avoid retrieval

- 10 errors related to the choice of aerosol model, we used one of the biomass burning aerosol model included in the fast algorithm. The particles effective radius is 0.15 microns and the single scattering albedo is equal 0.91 at 865 nm. The simulated 3D angular polarized reflectances as a function of the scattering angles are presented in Figure 6 (solid blue and red lines). Compared to the 1D reflectances with aerosols above cloud (dashed blue and red lines), the cloud heterogeneity effects amplify the
- 15 increase of the forward signal and the decrease of the cloudbow signal. As with molecular scattering (Section 4.1), aerosol scattering contributes to enhance the polarized reflectances in the shadow and cloud-free parts leading to higher averaged polarized reflectances in the forward direction. In the cloudbow direction (near 140°), °), and, to a lesser extent, in the side scattering (between 100° and 130° in scattering angle), the polarized reflectances are additionally attenuated because of the plane-parallel
- 20 biases. Note that for other solar zenith angles, the plots are similar with a more restricted scattering angular range (between 100° and 180° for SZA=20° and between 80° and 180° for SZA=40°), Consequently, only the attenuation due to the plane parallel bias impacts the measurements. The results obtained with the operational algorithm are presented in Table 3. We remind that the same input AOT is used in the 1D and 3D simulations (AOT of 0.15 at 865 nm). As expected, the AOTs
- 25 retrieved by the algorithm for homogenous clouds (1D input) are close to the input one, whatever the SZA value. The retrieved AOTs only slightly overestimate the input one (0.15) and are respectively equal to 0.18, 0.17, 0.17 for SZA of 20, 40 and 60°. This overestimation is likely due to the approximations used in the retrieval algorithm (e.g. interpolation of the LUTs). Comparing with the

| Supprimé:                   | reproduced                                               |
|-----------------------------|----------------------------------------------------------|
| Supprimé:                   | it can be limited                                        |
| Supprimé: (θ > 120°) | for only available data in the backscattering directions |
| Supprimé:                   | have no                                                  |

|----|---------------|

|------------------------------------------------------------|

|----|-----------------------------------------------------------------------------------------------------------------------------------------------------|

retrieved values from homogeneous cloud, significant departures are observed for fractional clouds (3D input) depending on the SZA. The AOTs retrieved at 865 nm are then equal to 0.119, 0.17 and 0.28 for SZA of 20, 40 and 60°, respectively. For a given solar zenith angle, the viewing geometries and the angular resolution are identical for the 1D and 3D. The differences observed in AOT between the 1D and 3D calculations are then necessarily due to 3D effects.

- The difference of AOT retrieval between 1D and 3D inputs depends on the solar zenith angle. Note that in the Table 3, the Ångström exponent is related to the ratio of two optical thicknesses at two wavelengths and corresponds in the retrieval to the best-selected model.
- For SZA=40°, the best model that minimized the cost function is the same for the homogeneous and
   fractional cloud. Differences for the retrieved AOT are negligible, but we note that the RMSE between the input and recalculated reflectances is slightly larger for the fractional cloud than for the homogenous one.

For  $SZA = 20^{\circ}$ , the operational algorithm also successfully retrieves the input aerosol model for the homogeneous and fractional cloud. However, the AOT retrieved by the operational algorithm, under the

- 15 1D assumption, is underestimated with error between -35 and -40%. For a SZA of 20°, the range of scattering angles effectively used for the retrieval is between 100° and 130°. Polarized reflectances for SZA=20° are not shown but they are similar to the ones shown in Figure 7 between 100 and 180°. Over the 100-130°, as shown in Figure 7, 3D polarized reflectances are lower than the 1D ones because of the plane-parallel biases, which explains why the AOT retrieved by the algorithm is underestimated.
- 20 However, as the differences are mainly due the plane-parallel bias, which is similar for the two wavelengths, the cloud heterogeneity effects do not affect the selection of the best aerosol model.

For SZA = 60°, the range of scattering angles used is between 60° and 130°. Between 60° and 90°, there is an increase of the forward scattering signal due to 3D effects, which is interpreted by the operational
algorithm as an increase in the AOT. We note also that 3D effects bias the aerosol model for this case as a smaller value of Ångström exponent (corresponding to a larger effective radius) is retrieved for the fractional cloud. The retrieved AOT is thus higher (AOT of 0.28 comparing to 0.17) with a relative

19

**His en forme :** Police :(Par défaut) Times New Roman, Non Italique, Anglais (E.U.) error up to 65%. For SZA=60°, the 3D effects consist in an increase of the polarized signal because of additional scattering in the clear sky parts. This increase is higher at 865 nm than at 670 nm. This leads to the selection by the algorithm of an erroneous model with a smaller Angström exponent.

Note that, in the operational algorithm, the algorithm is not applied for pixels too heterogeneous. 5 Those are filtered using the standard deviation of the COT retrieved at 1 km by MODIS that should not exceed 5. For the fractional cloud of this study, we checked the standard deviation value computed from the input cloud optical thickness (different from the retrieved one) and found 7. It is slightly above the homogeneity limit fixed in the aerosol above cloud algorithm developed for POLDER (Waquet et al., 2013). The results presented here for aerosol above cloud retrieval can thus be seen as an upper limit for

10 the operational algorithm,

**5. Conclusion**

This study used simulations to understand and quantify effects of cloud heterogeneities on POLDER total and polarized reflectances. We investigate the consequences of heterogeneous cloud radiative effects on the retrieved values of cloud optical thickness, droplet effective radius, effective variance,

15 cloud pressure and optical properties (optical thickness and Angstrom coefficient) of above cloud aerosol, provided by operational and research algorithms of the POLarization and Directionaly of Earth Reflectance (POLDER) instrument, 3D cloud fields were generated with the 3DCLOUD model (Szczap et al., 2014; Alkassem et al., 2017) and the 1D and 3D radiative transfer simulations were done with the Monte Carlo 3DMCPOL model (Cornet et al., 2010). Three types of heterogeneous water cloud were 20 studied: a flat, a bumpy and a fractional cloud.

The reflectances simulated at small spatial scale (50 m) and averaged at the POLDER spatial scale (7 km x 7 km) are used as realistic input of the different cloud operational and research algorithms. For high solar illumination (SZA=20°), the optical thickness retrieval yields, as it was already shown in numerous studies, lower optical thickness than the averaged ones because of the plane-parallel bias. For

25 POLDER, the retrieved optical thicknesses are underestimated by 10 or 35% depending on the cloud type. For oblique solar incidence, the POLDER algorithm yields higher optical thickness in the backscattering direction due to solar illumination effects and much lower optical thickness (up to -70%)

20

The Angström exponent is related to the ratio of two optical thicknesses at the two wavelengths and corresponds in the retrieval to the best-selected model. For SZA= $20^\circ$ , differences are mainly due the plane-parallel bias, which is similarclose for the two wavelengths. The ratio and Cconsequently, the selection of the best aerosol model the Angstrom exponent are thusare not affected by cloud heterogeneity effects.

[revised manuscript text omitted]

| observations and climate impacts. Rep. Prog. Phys. 73, 026801. https://doi.org/10.1088/0034-4885/73/2/026801       Mis enforme : Police ::11 pt.         Deschamps, PY., Breon, FM., Leroy, M., Podaire, A., Bricaud, A., Buriez, JC., 
[revised manuscript text omitted]
.                                                                                                                                                                                                                                                                                                                                                                                                                                                                                                                                                                                                                                                                                                                                                                                                                                                                                                                                                                                                                                                                                                                                                                                                                                                                                                                                                                                                                                                                                                                                     |  |
| 32